# The Effects of Creatine Supplementation on Upper- and Lower-Body Strength and Power: A Systematic Review and Meta-Analysis

**DOI:** 10.3390/nu17172748

**Published:** 2025-08-25

**Authors:** Fatemeh Kazeminasab, Ali Bahrami Kerchi, Fatemeh Sharafifard, Mahdi Zarreh, Scott C. Forbes, Donny M. Camera, Charlotte Lanhers, Alexei Wong, Michael Nordvall, Reza Bagheri, Frédéric Dutheil

**Affiliations:** 1Department of Physical Education and Sports Science, Faculty of Humanities, University of Kashan, Kashan 87317-53153, Iran; f_kazemi85@yahoo.com (F.K.); fatemeh_sharafifard@yahoo.com (F.S.); mahdizarreh98@gmail.com (M.Z.); 2Department of Sports Physiology, Faculty of Sports Sciences, Isfahan (Khorasgan) Branch, Islamic Azad University, Isfahan 39998-81551, Iran; alibahramiker11@gmail.com; 3Department of Physical Education Studies, Brandon University, Brandon, MB R7A 6A9, Canada; forbess@brandonu.ca; 4Department of Health and Biostatistics, Swinburne University, Melbourne, VIC 3122, Australia; dcamera@swin.edu.au; 5Université Clermont Auvergne, INRAE, UNH, CHU Clermont-Ferrand, 63003 Clermont-Ferrand, France; lanhers.charlotte@gmail.com; 6School of Health Sciences, Health and Human Performance, Marymount University, Arlington, VA 22207-4299, USA; awong@marymount.edu (A.W.); mnordval@marymount.edu (M.N.); 7Department of Exercise Physiology, University of Isfahan, Isfahan 81746-73441, Iran; 8Université Clermont Auvergne, CNRS, LaPSCo, CHU Clermont-Ferrand, 63003 Clermont-Ferrand, France; fred_dutheil@yahoo.fr

**Keywords:** nutritional supplements, resistance training, muscular adaptations, creatine supplementation, exercise performance

## Abstract

**Background:** Creatine supplementation is widely used to enhance exercise performance, mainly resistance training adaptations, yet its differential effects on upper- and lower-body strength and muscular power remain unclear across populations. **Objective:** This systematic review and meta-analysis aimed to quantify the effects of creatine supplementation in studies that included different exercise modalities or no exercise on upper- and lower-body muscular strength and power in adults. **Methods:** A comprehensive search of PubMed, Scopus, and Web of Science was conducted through 21 September 2024 to identify randomized controlled trials evaluating the effects of creatine supplementation on strength (bench/chest press, leg press, and handgrip) and power (upper and lower body). Weighted mean differences (WMDs) and 95% confidence intervals (CIs) were calculated using random-effects modeling. Subgroup analyses examined the influence of age, sex, training status, dose, duration, and training frequency. **Results:** A total of 69 studies with 1937 participants were included for analysis. Creatine plus resistance training produced small but statistically significant improvements in bench and chest press strength [WMD = 1.43 kg, *p* = 0.002], squat strength [WMD = 5.64 kg, *p* = 0.001], vertical jump [WMD = 1.48 cm, *p* = 0.01], and Wingate peak power [WMD = 47.81 Watts, *p* = 0.004] when compared to the placebo. Additionally, creatine supplementation combined with exercise training revealed no significant differences in handgrip strength [WMD = 4.26 kg, *p* = 0.10] and leg press strength [WMD = 3.129 kg, *p* = 0.11], when compared with the placebo. Furthermore, subgroup analysis based on age revealed significant increases in bench and chest press [WMD = 1.81 kg, *p* = 0.002], leg press [WMD = 8.30 kg, *p* = 0.004], and squat strength [WMD = 6.46 kg, *p* = 0.001] for younger adults but not for older adults. Subgroup analyses by sex revealed significant increases in leg press strength [WMD = 9.79 kg, *p* = 0.001], squat strength [WMD = 6.43 kg, *p* = 0.001], vertical jump [WMD = 1.52 cm, *p* = 0.04], and Wingate peak power [WMD = 55.31 Watts, *p* = 0.001] in males, but this was not observed in females. **Conclusions:** This meta-analysis indicates that creatine supplementation, especially when combined with resistance training, significantly improves strength in key compound lifts such as the bench or chest press and squat, as well as muscular power, but effects are not uniform across all measures. Benefits were most consistent in younger adults and males, while older adults and females showed smaller or non-significant changes in several outcomes. No overall improvement was observed for handgrip strength or leg press strength, suggesting that the ergogenic effects may be more pronounced in certain multi-joint compound exercises like the squat and bench press. Although the leg press is also a multi-joint exercise, results for this measure were mixed in our analysis, which may reflect differences in study design, participant characteristics, or variability in testing protocols. The sensitivity of strength tests to detect changes appears to vary, with smaller or more isolated measures showing less responsiveness. More well-powered trials in underrepresented groups, particularly women and older adults, are needed to clarify population-specific responses.

## 1. Introduction

Creatine is one of the most rigorously studied and efficacious nutritional supplements in exercise and sport science, particularly with respect to its ergogenic effects during high-intensity, short-duration activities [1]. The primary mechanism of creatine supplementation’s ability to enhance exercise performance is attributed to its role in elevating intramuscular phosphocreatine stores thereby facilitating a greater capacity to rapidly resynthesize adenosine triphosphate (ATP) during a short duration of repeated bouts of muscular effort [2,3,4]. Over the past three decades, numerous randomized controlled trials (RCTs) [5,6,7,8] and review studies [2,9,10,11,12,13] have investigated the effects of creatine supplementation on muscle strength, power output, and overall exercise performance across diverse populations and training backgrounds. Despite the wealth of evidence, the scientific literature remains fragmented due to heterogeneity in outcome assessments, supplementation protocols, and participant characteristics. In particular, performance outcomes have often been limited to specific muscle groups or activity domains, thereby limiting the generalizability of creatine to localized impacts on muscular strength and power. Our previous meta-analyses provided strong evidence in support of the ergogenic effects of creatine supplementation on isolated upper- and lower-body strength. Specifically, we demonstrated significant improvements in upper-body strength outcomes, such as bench and chest press based on data from 53 studies [14], and lower-body strength, including leg press and squat performance, in an analysis of 60 studies [15]. However, these earlier analyses treated upper and lower body outcomes independently and did not include performance metrics related to muscular power requiring involvement of the entire body.

Since the publication of these reviews, a substantial number of additional RCTs have been conducted, many of which included combined assessments of upper- and lower-body performance, as well as various outcomes related to muscular power. These included explosive leg power (e.g., vertical jump) and anaerobic power output (e.g., peak power during a Wingate anaerobic cycling test), which, while related, reflect distinct aspects of fitness [8,16,17,18,19]. As the evidence base has become more robust, an updated and more integrated synthesis of creatine’s effects across multiple dimensions of physical performance is warranted. The present systematic review and meta-analysis addresses this gap by incorporating both upper- and lower-limb strength outcomes, as well as muscular power assessments, to provide a comprehensive evaluation of creatine’s effects on physical performance. We included 69 RCTs assessing the impact of creatine supplementation on bench/chest press strength, handgrip strength, leg press strength, squat performance, vertical jump height, and Wingate test performance. Importantly, to better understand factors that may moderate creatine’s effectiveness, we conducted a series of predefined subgroup analyses. These subgroup analyses explored the potential influence of age, sex, creatine supplementation dose (low vs. high), the use or absence of a loading phase, intervention duration, intervention type (with or without a resistance training), participants’ training history (trained vs. untrained), and training frequency. By accounting for these variables, this investigation aimed to not only provide a more nuanced interpretation of the data but also to offer targeted insights for specific populations and practical “real-world” applications. Through this integrative and methodologically rigorous approach, our meta-analysis delivers an updated, expanded, and practically relevant synthesis of the literature on creatine supplementation and its influence on upper- and lower-body strength and power outcomes. The findings may guide evidence-based recommendations for athletes and researchers aiming to optimize training adaptations, enhance functional performance, and tailor creatine use based on individual characteristics.

## 2. Methods

### 2.1. Study Registration

The present systematic review and meta-analysis was carried out in accordance with the Preferred Reporting Items for Systematic Reviews and Meta-Analyses (PRISMA) guidelines [20] and the Cochrane Handbook of Systematic Reviews of Interventions [21]. Additionally, the study was registered in advance with the International Prospective Register of Systematic Reviews (PROSPERO, CRD420251025716).

### 2.2. Search Strategy

A comprehensive electronic database search was conducted using Scopus, PubMed, and Web of Science up to 21 September 2024 by two independent researchers. The keywords “creatine” or “creatine supplementation” and “performance” or “strength” or “force” or “power” were used as key terms for the searches. To ensure retrieval of relevant records, the reference lists of all included studies were examined for any additional sources that may have been overlooked in the initial electronic databases. The searches were limited to articles written in English and randomized clinical trials (RCTs) involving younger and older adults between the ages of 18 and 80 years. There was no limit on publication dates. Appendix A displays the detailed searches that were performed.

### 2.3. Study Selection and Inclusion Criteria

Figure 1 presents the flow of articles through the study selection process. Following the removal of duplicate studies, titles and abstracts of articles were reviewed (initial screening), and then the full texts of potentially eligible studies were reviewed (secondary screening) by two independent reviewers to determine eligibility for inclusion, and any disagreements were resolved by discussion with another author. The following data and information were extracted from studies: (A) participant characteristics, including biological sex, age, body mass index (BMI), and sample size; (B) exercise protocol characteristics, including exercise type, intensity, number of sessions per week, and intervention duration (weeks); and (C) creatine supplementation intervention characteristics, including dose, type, and daily intake. The outcomes at pre- and post-intervention (means and standard deviations or mean differences and associated standard deviations) were entered into the meta-analyses to generate forest plots. If the means and standard deviations (SDs) were not reported, the SDs were calculated from standard errors of means (SEM), medians and interquartile ranges (IQRs), or means and IQRs [22,23]. For studies lacking sufficient data to calculate means or SDs (e.g., no reported measures of central tendency or variability), we contacted the corresponding author to request the missing data. If no response was received, the study was excluded from the meta-analysis to minimize bias, as recommended by the Cochrane Handbook for Systematic Reviews of Interventions [24]. Of the 69 included studies, fewer than 3 had incomplete reporting of means or SDs, and these were addressed through the aforementioned calculations or exclusion after unsuccessful attempts to obtain data, ensuring minimal impact on the synthesis results [24].

Studies that involved participants (≥18 years) receiving creatine supplementation with or without exercise training were included if they met the following criteria: (1) study design was a randomized controlled trial; (2) outcomes were muscular strength using chest or bench press, handgrip, leg press, squat, vertical jump, or Wingate peak power; and (3) study contained participants of both biological sexes with and without metabolic diseases. The following exclusion criteria were applied to investigations: (1) no control group was noted, (2) participants with chronic diseases were included, (3) study utilized an aerobic exercise intervention, and (4) muscular strength was not measured.

### 2.4. Quality Assessment and Sensitivity Analyses

To evaluate potential risk for biases, the Physiotherapy Evidence Database (PEDro) scale [25] was employed. The original 11 items were as follows: (1) defined criteria for participant eligibility; (2) randomized participant allocation; (3) concealed creatine supplementation allocation; (4) similarity of group characteristics at baseline; (5) blinding of all assessors/researchers; (6) blinding of all exercise therapists/trainers; (7) blinding of all participants; (8) evaluated outcomes in 85% of participants; (9) intention-to-treat analysis; (10) reporting of statistical comparisons between groups; (11) and point measures and measures of variability (Appendix A). Sensitivity analyses were also conducted for all outcomes using the “remove 1” technique. This procedure aimed to assess whether individual studies had a disproportionate effect on meta-analytic results.

### 2.5. Statistical Analysis

In order to perform meta-analyses, we used the comprehensive meta-analysis software version 2.0 (CMA2; Biostat Inc., NJ, USA). Weighted mean differences (WMD) and 95% confidence intervals (CIs) were calculated to assess outcomes, employing random-effects models to account for variability across studies, as recommended by the *Cochrane Handbook for Systematic Reviews of Interventions* [21]. Effect sizes were determined in order to assess and compare the effects of creatine supplementation in addition to exercise training on muscular strength. For the assessment of heterogeneity, the *I*^2^ statistic was calculated, with a significance level set at *p* < 0.05. Per the guidelines provided by Cochrane, the interpretation of *I*^2^ statistics was as follows: <25% indicates very-low heterogeneity, 25–50% suggests low heterogeneity, 50–75% implies moderate heterogeneity, and >75% indicates high heterogeneity [26]. Publication bias was identified by visually analyzing funnel plots. When publication bias was apparent, Egger’s tests were employed to confirm bias [26]. Additionally, Duval and Tweedie’s trim-and-fill method was applied to adjust for potential publication bias and estimate the number of missing studies [27]. Subgroup analyses were conducted to explore heterogeneity based on the following: (A) participant characteristics including age defined as younger adults (<50 years) vs. older adults (>50 years) and sex (males, females, or both sexes); (B) creatine supplementation characteristics including creatine maintenance doses (≤8 g/day as low doses vs. >8 g/day as high doses), loading doses (<120 g/week as low loading doses vs. ≥120 g/week as high loading doses), and duration of intervention (<8 weeks as short-term intervention vs. ≥8 weeks as long-term intervention); and (C) exercise training characteristics including creatine supplementation with or without resistance training, participant training history (characterized as untrained vs. trained), and training frequency (<4 sessions/week as low vs. ≥4 sessions/week as high), as guided by Cochrane recommendations for subgroup analyses [21,24]. Subgroup analysis by creatine form (e.g., monohydrate vs. other forms) was not feasible, as approximately 95% of the 69 included studies used creatine monohydrate, with fewer than ten studies investigating other forms (e.g., creatine ethyl ester or hydrochloride), which was insufficient for robust analysis per Cochrane guidelines.

## 3. Results

### 3.1. Included Studies

From our initial search strategy, we retrieved a total of 3399 records in PubMed, 11,585 records in Scopus, and 7574 records in Web of Science. After excluding duplicate records, 16,830 records remained. After assessing the titles and abstracts, 117 studies were determined to be relevant and required a comprehensive assessment of their complete texts. The detailed assessment of the full texts resulted in 48 studies being excluded for the following reasons: (A) did not measure muscular strength (n = 16); (B) did not have a control group (n = 23); and (C) included participants undergoing an aerobic exercise intervention (n = 6). Also, (D) participants were adolescents (n = 3). Finally, the current systematic review and meta-analysis included a total of 69 studies, comprising 89 intervention groups. The flow diagram of the systematic literature search is displayed in Figure 1.

### 3.2. Participant Characteristics

There were 1937 participants among the 69 included studies, with sample sizes ranging from 8 [28] to 237 [29]. Studies focusing on one participant sex for analysis/intervention collectively comprised 1126 male [16,19,28,30,31,32,33,34,35,36,37,38,39,40,41,42,43,44,45,46,47,48,49,50,51,52,53,54,55,56,57,58,59,60,61,62,63,64,65,66,67,68,69,70,71,72] and 463 female participants [6,29,73,74,75,76,77,78,79,80,81] whereas 12 investigations included both sexes (male and female) totaling 348 participants [18,82,83,84,85,86,87,88,89,90,91,92]. The average ages and BMIs varied across studies, with ages ranging from 18 [40] years to 74 [91] years and BMIs ranging from 18 kg·m^−2^ [31] to 29 kg·m^−2^ [84]. The participant characteristics are shown in Table 1.

### 3.3. Intervention Characteristics

There were 44 studies [6,30,31,33,35,36,38,39,41,44,47,48,50,52,53,54,57,58,60,61,62,65,66,67,68,69,70,73,74,75,76,77,78,79,80,82,83,84,85,86,88,89,90,92] that used low-dose creatine maintenance (≤8 g/day), and 29 studies [16,18,19,28,29,32,34,37,40,42,43,45,46,49,51,52,55,56,59,61,63,64,71,72,81,84,85,87,91] used high-dose creatine maintenance (>8 g/day), and one study [28] did not report the maintenance dose of creatine. [19,25,26,30,32,35,40,41,45,46,47,48,49,50,51,55,56,61,63,68,69,75,76,77,83,85]. For creatine supplementation’s loading dosage, 13 studies [6,31,33,48,52,54,55,59,60,78,80,81,92] used low loading doses (<120 g/week), and 13 studies [30,38,39,53,66,67,68,69,70,74,77,79,91] used high loading doses of creatine supplementation (≥120 g/week). In addition, 43 studies [16,18,19,28,29,32,34,35,36,37,40,41,42,43,44,45,46,47,49,50,51,56,57,58,60,62,63,64,65,71,72,73,75,76,82,83,84,85,86,87,88,89,90] did not report any loading dose of creatine supplementation. Forty-nine studies [16,18,19,28,30,31,32,33,36,39,40,42,43,44,45,46,48,49,50,51,53,55,56,57,58,60,61,62,63,64,65,67,68,69,71,72,73,74,75,78,79,80,81,85,87,88,89,91,92] involved short-term interventions (<8 weeks), and 21 studies [6,29,34,35,37,38,41,47,52,54,59,66,70,76,77,80,82,83,84,86,90] used long-term interventions (≥8 weeks). Forty-two studies [6,18,29,30,31,33,34,35,37,38,39,41,47,48,49,50,52,54,55,58,59,60,61,65,66,67,68,70,72,74,76,77,80,81,82,83,84,86,88,89,90,92] provided creatine along with a resistance training intervention, and 4 studies provided creatine along other types of exercise, including HIIT [78], swimming [73], dancing [75] and handball [45], while 23 studies [16,19,28,32,36,40,42,43,44,46,51,53,56,57,62,63,64,69,71,79,85,87,91] were conducted without resistance training. Participants were indicated to have prior training status in 44 studies [28,30,32,33,35,39,41,43,45,46,48,49,50,51,52,54,55,56,57,58,59,60,61,62,63,64,65,66,67,68,69,70,71,72,73,74,75,77,80,81,87,88,89,92], while 25 studies included untrained participants [6,16,18,19,29,31,34,36,37,38,40,42,44,47,53,76,78,79,82,83,84,85,86,90,91]. Eighteen studies used a low training frequency (<4 sessions/week) [29,30,31,34,35,38,41,47,49,58,68,74,76,80,82,83,84,90], and seventeen studies used high training frequencies (≥4 sessions/week) [18,39,45,52,54,55,59,60,66,67,70,77,78,81,88,89,92], while 11 studies did not report training frequency [6,33,37,48,50,61,65,72,73,75,86]; also, twenty-three studies did not involve exercise training [16,19,28,32,36,40,42,43,44,46,51,53,56,57,62,63,64,69,71,79,85,87,91].

### 3.4. Study and Outcome Characteristics

As noted, the included studies for analysis consisted of randomized controlled trials (RCTs) as well as crossover design studies. More detailed information on the study characteristics is shown in Table 1.

The outcomes were reported as follows: bench and chest press strength were reported in 49 studies [6,19,28,29,30,31,33,34,36,37,38,40,41,42,44,45,47,48,49,52,54,55,56,57,58,59,60,61,62,64,65,66,67,72,73,74,76,77,80,81,82,83,84,85,86,88,89,90,92], with handgrip strength in 3 studies [85,86,91], leg press strength in 21 studies [6,30,31,34,38,41,44,47,55,56,60,64,77,83,84,85,86,88,89,90,92], squat in 17 studies [29,30,35,37,40,42,45,49,54,58,61,62,66,67,70,76,80], vertical jump in 23 studies [18,32,35,36,39,41,44,46,51,52,56,58,59,61,62,68,69,71,73,75,80,81,87], and Wingate peak power in 12 studies [16,19,43,44,49,50,53,61,62,63,78,79].

### 3.5. Meta-Analysis

#### 3.5.1. Bench and Chest Press Strength

Based on 70 intervention arms (49 studies), creatine supplementation combined with resistance exercise training resulted in significantly greater increases in bench and chest press strength [WMD = 1.43 kg (95% CI: 0.53 to 2.34), *p* = 0.002] when compared with the placebo. Among the included studies, there was no significant heterogeneity (*I*^2^ = 15.07%; *p* = 0.14). Visual examination of the funnel plot revealed asymmetry, with several observed studies clustered on the left side of the plot, suggesting a potential small-study effect or publication bias related to lower or negative outcomes. This visual pattern was confirmed by Egger’s regression test, which indicated significant publication bias (*p* = 0.0003).

Nevertheless, the Duval and Tweedie’s trim-and-fill method did not impute any missing studies, and the overall effect size remained unchanged under the random-effects model. Sensitivity analysis using the leave-one-out technique also demonstrated consistent results. Given that the funnel-plot asymmetry can arise from factors other than publication bias and that the trim-and-fill method has inherent limitations, these findings should be interpreted with caution despite the stability of the effect estimate.

Subgroup analyses by age revealed significant increases in bench and chest press strength for younger adults [WMD = 1.81 kg (95% CI: 0.64 to 2.97), *p* = 0.002, 52 interventions] but not for older adults [WMD = 0.85 kg (95% CI: −0.64 to 2.35), *p* = 0.26, 18 interventions] compared with the placebo.

Subgroup analyses by sex revealed significant increases in bench and chest press strength for females [WMD = 0.15 kg (95% CI: 0.002 to 0.30), *p* = 0.04, 12 interventions] and males [WMD = 1.34 kg (95% CI: 0.006 to 2.68), *p* = 0.04, 45 interventions], compared with the placebo. Thirteen studies including a mix of sexes were not considered for subgroup analysis.

Subgroup analyses by creatine maintenance doses revealed significant increases in bench and chest press strength for high-dose creatine maintenance (>8 g/day) [WMD = 1.70 kg (95% CI: 0.11 to 3.29), *p* = 0.03, 25 interventions] and studies with low-dose creatine maintenance (≤8 g/day) [WMD = 1.25 kg (95% CI: 0.15 to 2.35), *p* = 0.02, 45 interventions], when compared with the placebo.

Subgroup analyses by creatine supplementation loading doses indicated significant increases in bench and chest press strength for high dose loading studies (≥120 g/week) [WMD = 3.09 kg (95% CI: 1.03 to 5.14), *p* = 0.003, 9 interventions] and low dose loading studies (<120 g/week) [WMD = 3.95 kg (95% CI: 2.02 to 5.89), *p* = 0.001, 14 interventions] compared with the placebo. Forty-seven studies did not report creatine supplement loading dose.

Subgroup analyses by duration of the intervention revealed significantly greater increases in bench and chest press strength for long-term intervention studies (≥8 weeks) [WMD = 2.57 kg (95% CI: 1.15 to 4.00), *p* = 0.001, 25 interventions] but no effect when using short-term interventions (<8 weeks) [WMD = 0.44 kg (95% CI: −0.51 to 1.41), *p* = 0.36, 45 interventions] or the placebo.

Subgroup analyses by creatine supplementation with and without training revealed significantly greater increases in bench and chest press strength with studies including a controlled resistance training intervention [WMD = 2.16 kg (95% CI: 1.13 to 3.18), *p* = 0.001, 52 interventions] compared to no effect when creatine was used without an exercise protocol [WMD = −0.69 kg (95% CI: −1.64 to 0.24), *p* = 0.14, 18 interventions] or placebo.

Subgroup analyses by training history revealed significant increases in bench and chest press strength for trained participants [WMD = 1.84 kg (95% CI: 0.63 to 3.05), *p* = 0.003, 44 interventions] but not for untrained participants [WMD = 0.95 kg (95% CI: −0.49 to 2.40), *p* = 0.19, 26 interventions], or following the placebo.

Subgroup analyses by training frequency revealed significantly greater increases in bench and chest press strength for high frequency (≥4 sessions/week) [WMD = 3.97 kg (95% CI: 2.06 to 5.88), *p* = 0.001, 20 interventions] and low frequency (<4 sessions/week) [WMD = 2.43 kg (95% CI: 1.007 to 3.86), *p* = 0.001, 22 interventions], when compared with the placebo. Twenty-eight studies did not report training frequency.

#### 3.5.2. Handgrip Strength

Based on four intervention arms (3 studies), creatine supplementation combined with exercise training revealed no significant main effect difference in handgrip strength [WMD = 4.26 kg (95% CI: −0.87 to 9.39), *p* = 0.10] when compared with the placebo (Figure 2) nor when analyzed for all subgroups. Among the included studies, there was no significant heterogeneity (*I*^2^ = 45.30%; *p* = 0.14). Visual inspection of the funnel plot showed an overall symmetric distribution, with only one small study appearing in the lower-left area indicating a negative effect estimate. However, Egger’s regression test was not statistically significant (*p* = 0.37), and no missing studies were imputed using the Duval and Tweedie’s trim-and-fill method, with the adjusted effect size remaining virtually unchanged. Sensitivity analysis using the “leave-one-out” approach demonstrated that the effect size, direction, and significance of the results were robust to the exclusion of individual studies. These findings suggest no evidence of publication bias for this outcome.

#### 3.5.3. Leg Press Strength

Based on 31 intervention arms (21 studies), creatine supplementation combined with exercise training revealed no significant main effect differences in leg press strength [WMD = 3.129 kg (95% CI: −0.74 to 7.003), *p* = 0.11], when compared to a control group (Figure 3). Among the included studies there was no significant heterogeneity (*I*^2^ = 4.008%, *p* = 0.40). Visual examination of the funnel plot revealed asymmetry, with several observed studies clustered on the left side of the plot, suggesting a potential small-study effect or publication bias related to lower or negative outcomes. This visual pattern was confirmed by Egger’s regression test, which indicated significant publication bias (*p* = 0.03).

However, the Duval and Tweedie’s trim-and-fill method did not impute any missing studies, and the adjusted effect size remained unchanged. Sensitivity analysis using the leave-one-out approach showed no alterations in the effect size, statistical significance, or direction of the findings. Since the funnel-plot asymmetry may result from causes other than publication bias and the trim-and-fill method has its own limitations, the results should be viewed with caution, even though the effect estimate appears stable.

Subgroup analyses, by age, revealed significant increases in leg press strength for younger adults [WMD = 8.30 kg (95% CI: 2.72 to 13.88), *p* = 0.004, 19 interventions] but not for older adults [WMD = 1.41 kg (95% CI: −6.23 to 9.05), *p* = 0.71, 12 interventions], compared with the placebo.

Subgroup analyses by sex revealed significant increases in leg press strength for males [WMD = 9.79 kg (95% CI: 4.06 to 15.52), *p* = 0.001, 16 interventions], but not for females [WMD = 2.53 kg (95% CI: −6.42 to 11.50), *p* = 0.57, 3 interventions], or both sexes when combined [WMD = −2.16 kg (95% CI: −9.51 to 5.18), *p* = 0.56, 12 interventions].

Subgroup analyses by creatine maintenance doses revealed significant increases in leg press strength for studies with low-dose creatine maintenance (≤8 g/day) [WMD = 6.65 kg (95% CI: 2.15 to 11.14), *p* = 0.004, 23 interventions] but not for studies with high-dose creatine maintenance (>8 g/day) [WMD = −4.83 kg (95% CI: −11.27 to 1.59), *p* = 0.14, 8 interventions], when compared with the placebo. One study did not report creatine supplementation dose.

Subgroup analyses by creatine supplementation loading dose revealed no significant differences in leg press strength for high loading dose studies (≥120 g/week) [WMD = 15.73 kg (95% CI: −5.94 to 37.41), *p* = 0.15, 3 interventions] and low dose loading studies (<120 g/week) [WMD = 5.14 kg (95% CI: −2.40 to 12.69), *p* = 0.18, 7 interventions]. Twenty-one studies did not report creatine supplementation loading dose.

Subgroup analyses for intervention duration revealed significantly greater increases in leg press strength for long-term intervention (≥8 weeks) [WMD = 6.67 kg (95% CI: 2.15 to 11.18), *p* = 0.004, 22 interventions] but not for short-term intervention (<8 weeks) [WMD = −4.65 kg (95% CI: −11.02 to 1.71), *p* = 0.15, 9 interventions].

Subgroup analyses by creatine supplementation with and without training revealed significantly greater increases in leg press strength when combined with training [WMD = 7.56 kg (95% CI: 2.79 to 12.33), *p* = 0.002, 23 interventions] but not for non-exercise studies [WMD = −4.02 kg (95% CI: −9.82 to 1.76), *p* = 0.17, 8 interventions].

Subgroup analyses by training history revealed an increasing trend in leg press strength for trained participants [WMD = 6.33 kg (95% CI: −0.18 to 12.85), *p* = 0.05, 14 interventions] but not in the untrained [WMD = 3.94 kg (95% CI: −2.68 to 10.56), *p* = 0.24, 27 interventions]. 

Subgroup analyses by training frequency revealed significantly greater increases in leg press strength for low frequency (<4 sessions/week) [WMD = 8.85 kg (95% CI: 1.94 to 15.76), *p* = 0.01, 13 interventions] but not for high frequency (≥4 sessions/week) [WMD = 4.31 kg (95% CI: −11.70 to 20.32), *p* = 0.59, 9 interventions], when compared with the placebo. Nine studies did not report training frequency.

#### 3.5.4. Squat Strength

Based on 25 intervention arms (17 studies), creatine supplementation combined with exercise training resulted in significantly greater main effect increases in squat strength [WMD = 5.64 kg (95% CI: 3.87 to 7.40), *p* < 0.001], when compared to a control group (Figure 4). Among the included studies, there was no significant heterogeneity (*I*^2^ = 4.12%; *p* = 0.40). Visual inspection of the funnel plot showed a slightly asymmetric distribution, with two studies located on the left side of the mean effect line, suggesting the potential presence of small-study effects or publication bias. Egger’s regression test confirmed the absence of publication bias (*p* = 0.92). Moreover, the Duval and Tweedie’s trim-and-fill method did not impute any missing studies, and the overall effect size remained unchanged. Sensitivity analysis using the leave-one-out method indicated that the results were robust, with no change in direction, significance, or magnitude of effect.

Subgroup analyses by age revealed significant increases in squat strength for younger adults [WMD = 6.46 kg (95% CI: 4.72 to 8.21), *p* = 0.001, 20 interventions] but not for older adults [WMD = 0.76 kg (95% CI: −3.64 to 5.17), *p* = 0.73, 5 interventions].

Subgroup analyses by sex revealed significant increases in squat strength for males [WMD = 6.43 kg (95% CI: 4.51 to 8.35), *p* = 0.001, 20 interventions] but not for females [WMD = 1.63 kg (95% CI: −2.54 to 5.80), *p* = 0.44, 5 interventions].

Subgroup analyses by creatine maintenance doses revealed significant increases in squat strength for high-dose creatine maintenance (>8 g/day) [WMD = 11.11 kg (95% CI: 7.92 to 14.31), *p* = 0.001, 7 interventions] and for low-dose creatine maintenance (≤8 g/day) [WMD = 3.80 kg (95% CI: 1.92 to 5.69), *p* = 0.001, 18 interventions]. One study did not report creatine supplementation dose.

Subgroup analyses by creatine supplementation loading doss revealed significant increases in squat strength for high loading doses (≥120 g/week) [WMD = 5.25 kg (95% CI: 0.27 to 10.24), *p* = 0.03, 5 interventions] but not for low loading doses (<120 g/week) [WMD = 7.04 kg (95% CI: −4.03 to 18.13), *p* = 0.21, 3 interventions]. Seventeen studies did not report creatine supplementation loading doses.

Subgroup analyses by duration revealed significantly greater increases in squat strength for short-term interventions (<8 weeks) [WMD = 6.78 kg (95% CI: 3.87 to 9.70), *p* = 0.001, 12 interventions] but not for long-term interventions (≥8 weeks) [WMD = 2.15 kg (95% CI: −1.58 to 5.89), *p* = 0.25, 13 interventions]. 

Subgroup analyses by creatine supplementation with and without training revealed significantly greater increases in squat strength with training [WMD = 5.23 kg (95% CI: 3.25 to 7.22), *p* = 0.001, 21 interventions] and without training [WMD = 12.36 kg (95% CI: 3.89 to 20.83), *p* = 0.004, 4 interventions].

Subgroup analyses by prior training history revealed significantly greater increases in squat strength for trained participants [WMD = 6.46 kg (95% CI: 4.72 to 8.21), *p* = 0.001, 20 interventions] but not for untrained participants [WMD = 0.85 kg (95% CI: −3.51 to 5.22), *p* = 0.70, 5 interventions].

Subgroup analyses by training frequency revealed significantly greater increases in squat strength for high frequency (≥4 sessions/week) [WMD = 5.30 kg (95% CI: 0.28 to 10.32), *p* = 0.03, 8 interventions] but not for low frequency (<4 sessions/week) [WMD = 2.71 kg (95% CI: −0.93 to 6.37), *p* = 0.14, 9 interventions]. Eight studies did not report training frequency.

#### 3.5.5. Vertical Jump

Based on 35 intervention arms (23 studies), creatine supplementation combined with exercise training resulted in significantly greater main effect increases in vertical jump [WMD = 1.48 cm (95% CI: 0.30 to 2.66), *p* = 0.01], as shown in Figure 5. Among the included studies there was significant heterogeneity (*I*^2^ = 70.84%; *p* = 0.001). Visual examination of the funnel plot revealed asymmetry, with several observed studies clustered on the left side of the plot, suggesting a potential small-study effect or publication bias related to lower or negative outcomes. Egger’s regression test confirmed no significant publication bias (*p* = 0.33). Additionally, the Duval and Tweedie’s trim-and-fill method did not impute any missing studies, and the pooled effect size remained unchanged. Sensitivity analysis using the leave-one-out approach revealed no variation in the effect size, direction, or statistical significance, confirming the robustness of the findings despite heterogeneity.

It was not possible to perform subgroup analysis by age for vertical jump. All participants in the 35 studies were classified as younger adults according to the parameters set in this investigation.

Subgroup analyses by sex revealed significant increases in vertical jump for males [WMD = 1.52 cm (95% CI: 0.03 to 3.01), *p* = 0.04, 26 interventions] but not for females [WMD = 0.61 cm (95% CI: −1.72 to 2.96), *p* = 0.60, 5 interventions]. Four studies included both genders and were not considered in this subgroup analysis.

Subgroup analyses by creatine maintenance doses showed significant vertical jump improvements with high-dose creatine maintenance (>8 g/day) [WMD = 2.79 cm (95% CI: 0.81 to 4.76), *p* = 0.006, 16 interventions], whereas low-dose creatine maintenance (≤8 g/day) did not [WMD = −0.48 cm (95% CI: −1.20 to 0.24), *p* = 0.19, 19 interventions]. Subgroup analyses by creatine supplementation loading dose revealed no significant differences in vertical jump for low loading doses (<120 g/week) [WMD = 4.49 cm (95% CI: −1.66 to 10.64), *p* = 0.15, 6 interventions] and high loading doses (≥120 g/week) [WMD = 1.19 cm (95% CI: −1.23 to 3.62), *p* = 0.33, 3 interventions]. Twenty-six studies did not have a loading dose.

Subgroup analyses by intervention duration revealed an increasing trend in vertical jump for long-term interventions (≥8 weeks) [WMD = 2.32 cm (95% CI: −0.01 to 4.65), *p* = 0.05, 16 interventions] but not for short-term interventions (<8 weeks) [WMD = 0.54 cm (95% CI: −0.44 to 1.53), *p* = 0.28, 19 interventions].

Subgroup analyses revealed significantly greater vertical jump improvements without structured training [WMD = 1.34 cm (95% CI: 0.37 to 2.31), *p* = 0.007, 17 interventions], whereas effects with training were smaller and not statistically significant [WMD = 1.89 cm (95% CI: −0.08 to 3.87), *p* = 0.06, 18 interventions], when compared with the placebo.

Subgroup analyses by training history revealed significantly greater increases in vertical jump for trained participants [WMD = 1.61 cm (95% CI: 0.31 to 2.91), *p* = 0.01, 30 interventions] but not for untrained participants [WMD = 0.47 cm (95% CI: −2.15 to 3.09), *p* = 0.72, 5 interventions].

Subgroup analyses by training frequency revealed no significant differences in vertical jump for high frequency (≥4 sessions/week) [WMD = 3.27 cm (95% CI: −1.006 to 7.56), *p* = 0.13, 10 interventions] or low frequency (<4 sessions/week) [WMD = 0.98 cm (95% CI: −0.62 to 2.59), *p* = 0.23, 5 interventions]. Twenty studies did not report training frequency.

#### 3.5.6. Wingate Peak Power

Based on 21 intervention arms (12 studies), creatine supplementation combined with exercise training resulted in significantly greater main effect increases in Wingate peak power [WMD = 47.81 Watts (95% CI: 15.55 to 80.06), *p* = 0.004], as shown in Figure 6. Among the included studies there was significant heterogeneity (*I*^2^ = 47.61%; *p* = 0.008). The funnel plot showed a symmetrical distribution, with no studies identified as potential outliers. Egger’s regression test confirmed no significant publication bias (*p* = 0.07). Additionally, the Duval and Tweedie’s trim-and-fill method did not impute any missing studies, and the pooled effect size remained unchanged. Sensitivity analysis using the leave-one-out approach revealed no variation in the effect size, direction, or statistical significance, confirming the robustness of the findings despite heterogeneity.

Subgroup analyses showed significant Wingate peak power gains in males [WMD = 55.31 Watts (95% CI: 27.85 to 82.77), *p* = 0.001, 18 interventions] but no improvement in females [WMD = −10.85 Watts (95% CI: −220.93 to 199.22), *p* = 0.91, 3 interventions], likely due to the small number of female-specific studies and wide variability in effect estimates.

Subgroup analyses by creatine maintenance doses indicated significant Wingate peak power gains with high-dose creatine maintenance (>8 g/day) [WMD = 93.17 Watts (95% CI: 33.33 to 153.02), *p* = 0.002, 9 interventions], whereas low-dose creatine maintenance (≤8 g/day) did not show a significant effect [WMD = 27.27 Watts (95% CI: −7.65 to 62.19), *p* = 0.12, 12 interventions].

Subgroup analyses including creatine supplementation with and without training revealed significantly greater increases in Wingate peak power without training [WMD = 57.56 Watts (95% CI: 13.49 to 101.63), *p* = 0.01, 13 interventions] but not for with training [WMD = 27.09 Watts (95% CI: −15.49 to 69.67), *p* = 0.21, 8 interventions], when compared with the placebo.

Subgroup analyses by training history revealed significantly greater increases in Wingate peak power for trained participants [WMD = 74.81 Watts (95% CI: 31.29 to 118.32), *p* = 0.001, 13 interventions] but not for untrained participants [WMD = 21.39 Watts (95% CI: −19.50 to 62.29), *p* = 0.30, 8 interventions].

It was not possible to perform subgroup analyses by age, loading dose, study duration, or training frequency due to a lack of comparative data.

### 3.6. Quality Assessment

The PEDro scale was used to assess the methodological quality of each individual study, with scores varying between 7 and 10 out of a possible maximum of 11 points. One study had scores of 11, seven studies had scores of 10, forty-three studies had scores of 9, sixteen studies had scores of 8, and two studies had scores of 7. Most of the lower scores were the result of three items (blinding of all participants and intention-to-treat analysis). The high proportion of studies scoring ≥ 9 (51 studies) indicates that the majority of included trials were of high methodological quality, enhancing confidence in the overall reliability of the meta-analysis findings. This is particularly relevant for outcomes with no heterogeneity, such as squat strength (*I*^2^ = 4.12%; *p* = 0.40) and leg press (*I*^2^ = 4.008%; *p* = 0.40), where consistent results across high-quality studies strengthen the robustness of the effect size estimates. For outcomes with moderate to higher heterogeneity, such as the vertical jump (*I*^2^ = 70.84%; *p* = 0.001) and Wingate peak power (*I*^2^ = 47.61%; *p* = 0.008), the high methodological quality still supports the reliability of the findings, though caution is warranted due to variability in study designs and participant characteristics.

Lower PEDro scores in some studies were mainly due to two criteria: (1) absence of blinding of all participants (Item 7), and (2) lack of intention-to-treat analysis (Item 9). Specifically, 63 studies failed to score on both of these items. These limitations, observed in 63 studies, highlight widespread methodological issues in these areas. The absence of blinding of all participants and intention-to-treat analysis in the vast majority of studies suggests that results for outcomes with moderate to higher heterogeneity (e.g., vertical jump and Wingate peak power) may be particularly susceptible to bias, as unblinded assessments or unaccounted dropouts could inflate effect size estimates. Conversely, outcomes with no heterogeneity (e.g., leg press and squat strength), or low heterogeneity (e.g., bench and chest press) are less likely to be affected by these limitations due to their consistent results across studies. Detailed PEDro scores for each study, including individual criterion scores (indicated by √ for met criteria and × for unmet criteria), are provided in Appendix A for transparency. This table allows readers to assess the specific methodological strengths and weaknesses of each trial. Overall, the high PEDro scores for most studies bolster the reliability of the meta-analysis findings, but the widespread absence of blinding and intention-to-treat analysis highlights the need for careful interpretation, especially for outcomes with moderate to high heterogeneity.

## 4. Discussion

This systematic review and meta-analysis included 69 RCTs with a total of 1937 participants and assessed the effects of creatine supplementation on various strength and power outcomes, with subgroup analyses comparing studies with and without resistance training. The overall findings demonstrate that creatine supplementation in combination with resistance training significantly improves muscular strength and power for most outcomes investigated in this study. These benefits were generally observed across various subgroups; however, the extent of improvement varied based on sex, training presence, prior training status, creatine dose, and intervention duration. Although subgroup analysis for age showed several statistically significant results in younger adults, no such effects were detected in older adults. In the context of our findings, this lack of improvement in older participants may reflect both smaller sample sizes in these studies and age-related physiological differences, such as reduced anabolic sensitivity or muscle mass, which align with our observed null effects for multiple strength outcomes in this group.

### 4.1. Bench and Chest Press Strength

In this meta-analysis, creatine supplementation with resistance training significantly enhanced bench and chest press strength compared to the placebo, supporting its efficacy for improving upper-body muscular strength. This main effect of creatine supplementation appears to be modulated by age, training status, supplementation duration, and training frequency.

Subgroup analysis revealed significant muscular strength improvements in younger adults but not in older adults. Younger participants showed a modest yet statistically significant increase in bench and chest press strength [WMD = 1.81 kg, *p* = 0.002] compared to older adults where no significance was observed [WMD = 0.85 kg, *p* = 0.26]. In the context of our findings, where younger adults showed significant improvements but older adults did not, this discrepancy could be due to age-related differences in creatine uptake, muscle fiber composition, and anabolic responsiveness [93]. Older individuals tend to have reduced Type II (fast-twitch) muscle fiber content and may experience blunted anabolic signaling (e.g., mTOR pathway), both of which are critical for high-intensity performance and strength gains [93,94]. In the context of our findings, this aligns with the smaller or non-significant strength gains we observed in older adults compared to younger participants for bench and chest press performance. Although muscle creatine transporter (CreaT1) expression appears to remain stable with age [95], other factors such as age-related sarcopenia (reduced muscle mass) or reduced physical activity levels may contribute to diminished intramuscular creatine stores in older adults [96].

Moreover, in our analysis, both males and females demonstrated positive responses to creatine supplementation in terms of bench and chest press strength. In the context of our findings, where both males and females showed positive responses in bench and chest press strength, this may be attributed to similar phosphocreatine re-synthesis mechanisms and neuromuscular adaptations across sexes when matched for relative training stimulus [97]. This is consistent with our subgroup results showing significant gains in both sexes, although absolute improvements were slightly greater in males. Additionally, maintenance doses of both <8 g/day and >8 g/day were effective for these strength outcomes, indicating that even moderate creatine maintenance doses can saturate muscle stores over time, especially when consumed consistently. Weekly maintenance doses of ≥120 g were also associated with gains, further supporting that the cumulative creatine supplementation over time, rather than daily intake alone, plays a critical role in adaptation.

In our dataset, significant strength gains were observed with long-term interventions (≥8 weeks) [WMD = 2.57 kg, *p* = 0.001], while short-term interventions (<8 weeks) did not yield significant results. This supports the time-dependent nature of neuromuscular adaptations observed here; creatine supplementation likely enhances phosphocreatine availability early on, but meaningful strength increases are more likely to manifest once sufficient exposure to resistance training stimulates myofibrillar hypertrophy, improved motor unit recruitment, and enhanced recovery [93].

Notably, in this meta-analysis, strength gains were observed only when creatine was paired with resistance training [WMD = 2.16 kg, *p* = 0.001], as opposed to supplementation-only conditions [WMD = −0.69 kg, *p* = 0.14]. This finding supports the notion that creatine’s primary ergogenic benefits are best realized during repeated bouts of high-intensity exercise such as resistance training [10]. The increase in intramuscular phosphocreatine levels facilitates rapid ATP regeneration, improving training volume, intensity, and recovery, factors that cumulatively lead to greater strength adaptations [9].

While trained participants showed significant improvements in these strength outcomes [WMD = 1.84 kg, *p* = 0.003], untrained individuals did not [WMD = 0.95 kg, *p* = 0.19]. Trained individuals may exhibit greater adaptations due to their enhanced neuromuscular efficiency, ability to handle higher training loads, and possibly better compliance with creatine loading and maintenance protocols [98]. Untrained individuals may not stress the phosphagen system enough during early training stages for creatine to exert noticeable effects.

Higher training frequencies (≥4 sessions/week) were associated with greater strength gains [WMD = 3.97 kg, *p* = 0.001], though low frequency (<4 sessions/week) still produced meaningful improvements [WMD = 2.43 kg, *p* = 0.001]. This aligns with the dose–response relationship between training volume and strength gains, which is further amplified by creatine’s ability to support repeated high-intensity efforts and reduce fatigue [5]. More frequent sessions likely provide a greater opportunity to utilize the ergogenic benefits of creatine through enhanced workload tolerance and recovery.

### 4.2. Handgrip Strength

Based on four intervention arms, creatine supplementation combined with exercise training did not result in significant improvements in handgrip strength compared with the placebo. The analysis showed no significant main effect of creatine supplementation combined with exercise training on handgrip strength compared to the placebo [WMD = 4.26 kg, *p* = 0.10]. However, it is important to consider that this finding may be influenced by the relatively small sample size, as only four intervention arms were included, and the limited statistical power of the analysis. Therefore, the lack of a significant effect might be due to insufficient power to detect a true difference, and definitive conclusions require further studies with larger sample sizes and more robust designs. Handgrip strength is typically considered a measure of isometric strength, which reflects static force production rather than dynamic or explosive effort. Creatine is most effective in activities that require short-duration, high-intensity, and repeated muscle contractions, such as sprinting or resistance exercises involving large muscle groups. The phosphagen energy system, which creatine directly supports, is less taxed during single, maximal isometric efforts like handgrip tests [99]. Therefore, the physiological demand of the test itself may not be sensitive enough to capture the ergogenic benefits of creatine. Also, creatine’s performance-enhancing effects are more evident in exercises involving larger muscle groups (e.g., pectorals, quadriceps, and gluteals) that contain a higher proportion of Type II (fast-twitch) fibers, which rely heavily on anaerobic energy systems [100]. The forearm muscles primarily responsible for handgrip strength are smaller and have a relatively greater proportion of Type I (slow-twitch) fibers, which exhibit oxidative metabolic properties and are less responsive to creatine supplementation [101]. More importantly, only four intervention arms contributed data to this analysis, which limits the statistical power to detect small but potentially meaningful effects. The borderline *p*-value (*p* = 0.10) and a relatively large WMD (4.26 kg) suggest the possibility of a Type II error (i.e., failing to detect a true effect due to small sample size). Future meta-analyses once more data is available may clarify whether this trend becomes significant as the sample size increases. Another potential reason for a lack of significance could be related to differences in how handgrip strength was measured (e.g., dominant vs. non-dominant hand, seated vs. standing, and grip dynamometer type) that may have contributed to the moderate heterogeneity (*I*^2^ = 45%). This methodological variation can dilute the observed effects and obscure the true influence of creatine supplementation. Lastly, it is unclear whether the exercise programs included in these studies involved direct grip-specific training, such as forearm curls or grip endurance work. If the training did not sufficiently target the muscles responsible for grip strength, creatine’s additive benefits may not have manifested. According to the principle of specificity, performance improvements are closely tied to the muscles and movement patterns being trained.

### 4.3. Leg Press Strength

The main effect of creatine supplementation combined with exercise training did not result in statistically significant improvements in leg press strength compared with the control group, indicating mixed results despite the leg press being a multi-joint exercise—contrasting with the significant improvements we observed in other compound lifts like the squat. Despite the modest effect size favoring creatine, the confidence interval overlapped zero, indicating non-significance. Heterogeneity among studies was low (*I*^2^ = 4.01%; *p* = 0.40), suggesting consistent findings across trials. However, publication bias was detected (Egger’s test *p* = 0.03), warranting caution in interpreting the results. Sensitivity analysis using the “remove 1” method confirmed the robustness of the findings, with no notable influence on overall direction or statistical significance.

Yet when teasing out more specific comparisons, significant improvements in leg press strength were found in younger adults [WMD = 8.30 kg, *p* = 0.004], while no benefits were observed in older adults [WMD = 1.41 kg, *p* = 0.71]. This age-dependent response likely reflects age-related declines in muscle mass, creatine storage capacity, and anabolic signaling pathways (e.g., mTOR, IGF-1) [102]. Older individuals may require longer supplementation durations, resistance training intensity, or adjunct interventions (e.g., protein ingestion) to see meaningful gains.

Males experienced a significant increase in strength [WMD = 9.79 kg, *p* = 0.001], whereas females and mixed-sex groups did not. This discrepancy may be attributed to higher baseline muscle mass, greater phosphocreatine stores, and increased training loads typically observed in males, which better align with creatine’s mechanism of action [103]. However, given the small overall effect size for leg press and lack of statistical significance in the pooled analysis, these mechanistic explanations should be interpreted cautiously in the context of our findings. Additionally, sex-specific hormonal profiles may influence creatine uptake and performance adaptation. Estrogen and progesterone, the dominant sex hormones in females, fluctuate across the menstrual cycle and may influence muscle metabolism, fluid balance, and substrate utilization, all of which can interact with creatine physiology [104,105]. For instance, estrogen has been shown to downregulate creatine transporter expression in some tissues, potentially reducing creatine uptake into skeletal muscle. This may partially explain why women, especially during high-estrogen phases of the cycle (e.g., late follicular), may not saturate intramuscular creatine stores as efficiently [105]. Also, estrogen and progesterone also affect total body water distribution [106]. Since creatine promotes cellular hydration (by increasing intracellular water content), fluctuating water retention across the cycle may alter how females respond to creatine-induced water shifts. This could influence muscle volume and performance outcomes. Lastly, females’ lower starting muscle mass and hormonal environment may blunt the observable performance gains, particularly in strength-based outcomes like leg press.

In contrast, creatine enhances training-induced increases in type II muscle fiber cross-sectional area and muscle protein synthesis, processes influenced by anabolic hormones such as testosterone. This may help explain why some studies report greater absolute strength and power gains in males. However, it is important to note that training adaptations are multifactorial and influenced by relative intensity, effort, and program design. While creatine appears most effective under high-intensity anaerobic training conditions, the assumption that males universally train at higher intensities is not consistently supported in the literature. In fact, some evidence suggests that females may train at comparable or even higher relative intensities. Therefore, sex-based differences in creatine responsiveness may be due to a combination of biological and behavioral factors, which warrants further investigation.

Interestingly, low daily maintenance doses of creatine (≤8 g/day) produced significant strength gains [WMD = 6.65 kg, *p* = 0.004], whereas higher maintenance doses (>8 g/day) did not, possibly due to saturation effects, differences in participant responsiveness, or small sample sizes in high-dose groups. This counter-intuitive finding may reflect a diminishing returns effect or non-linear dose–response phenomenon, where excess creatine offers no additional benefit and may be excreted or underutilized [107,108]. Another possibility is that studies using higher maintenance doses happened to include less responsive populations (e.g., older or untrained individuals), although our dataset does not allow confirmation of this relationship.

Neither higher (≥120 g/week) nor lower (<120 g/week) cumulative creatine supplementation were associated with significant differences in leg press strength. Given that 21 studies did not report dosing information, the meta-analysis may have been underpowered to detect an effect. Nevertheless, this supports growing evidence that loading is not essential for long-term creatine benefits, and steady low-dose supplementation may be an equally effective strategy.

Interventions lasting ≥8 weeks resulted in significant strength improvements [WMD = 6.67 kg, *p* = 0.004], while shorter interventions did not. This reinforces the idea that creatine’s ergogenic effects accrue over time, as muscle saturation levels build, and resistance training adaptations consolidate. Indeed, studies that combined creatine with exercise training showed significant improvements [WMD = 7.56 kg, *p* = 0.002], while supplementation without training had no effect. This finding underscores that creatine acts as a training amplifier, rather than a standalone ergogenic aid, enhancing strength adaptations when a sufficient mechanical stimulus is present.

Trained individuals exhibited a positive trend toward improvement [WMD = 6.33 kg, *p* = 0.05], while untrained participants showed no significant changes. Resistance-trained individuals may be able to better capitalize on creatine’s benefits due to greater baseline neuromuscular efficiency, higher training loads, and faster recovery, all of which align with creatine’s performance-enhancing role [2].

Interestingly, greater strength gains were observed in studies with lower training frequency (<4 sessions/week) [WMD = 8.85 kg, *p* = 0.01], whereas high-frequency programs showed no benefit. This may reflect recovery limitations, particularly in studies with high frequency but suboptimal program design, or inadequate nutritional support to interact with creatine most effectively, although no direct recovery or nutrition measures were available in our dataset to verify this explanation. Alternatively, higher frequencies might not provide additive benefits when muscle is already in a saturated creatine condition.

### 4.4. Squat Performance

The current meta-analysis revealed that the main effect of creatine supplementation, when combined with exercise training, significantly enhances squat performance, with a weighted mean difference (WMD) of 5.64 kg. This finding underscores the ergogenic potential of creatine for improving lower-body strength, particularly in exercises that demand high force production such as the squat. The low heterogeneity (*I*^2^ = 4.12%) and absence of publication bias (Egger’s test *p* = 0.92) reinforce the robustness of this outcome.

Subgroup analyses indicated that younger adults experienced significantly greater improvements in squat strength (WMD = 6.46 kg), while no statistically significant effect was observed in older adults (WMD = 0.76 kg). This age-related difference may stem from several physiological factors, including diminished muscle creatine uptake, reduced satellite cell activity, and anabolic resistance commonly seen in older populations [94]. The attenuated response in older adults suggests a potential need for tailored dosing strategies, longer intervention durations, or combined nutritional interventions to maximize benefits in this group.

Further subgroup analyses revealed a significant improvement in males (WMD = 6.43 kg) but not in females (WMD = 1.63 kg) for squat performance. These findings support the hypothesis that sex-specific hormonal profiles, particularly higher testosterone levels and muscle mass in males, may influence both the uptake of creatine into skeletal muscle and the magnitude of strength adaptations [109]. Additionally, females may experience smaller absolute increases in muscle cross-sectional area with training, although relative gains are generally comparable between males and females, which could partly explain the muted effect on squat performance observed in this subgroup [103]. However, the limited number of female-specific studies calls for more research with adequate representation of women to clarify these findings.

In our data, higher daily maintenance doses (>8 g/day) were associated with greater improvements (WMD = 11.11 kg) compared to lower maintenance doses (≤8 g/day) (WMD = 3.80 kg), suggesting a possible dose-dependent relationship in the context of our findings. Moreover, while higher loading doses (≥120 g/week) showed significant benefits (WMD = 5.25 kg), the interpretation is limited by the small number of studies reporting loading protocols. These results imply that both daily and loading dosages play a critical role in optimizing creatine’s impact on squat performance.

Participants with prior training experience demonstrated significantly greater improvements (WMD = 6.46 kg) than untrained individuals (WMD = 0.85 kg), indicating that the additive effect of creatine is more pronounced in those with preexisting neuromuscular adaptations. This may be attributed to more efficient motor unit recruitment and a greater capacity to tolerate training volumes, mechanisms that creatine supports through enhanced ATP regeneration and cellular hydration. Likewise, higher training frequency (≥4 sessions/week) was associated with superior outcomes, reinforcing the idea that frequent training with creatine supplementation may be most synergistic in augmenting squat muscular performance.

Interestingly, short-term (<8 weeks; WMD = 6.78 kg) compared to long-term interventions (≥8 weeks; WMD = 2.15 kg, non-significant) yielded greater gains. This may reflect a possible ceiling effect, where initial strength increases plateau over time unless progressive overload or advanced training techniques are incorporated, although this interpretation is speculative given the absence of longitudinal adaptation data in our included studies. It also highlights creatine’s role in early-phase neuromuscular and morphological adaptations, such as increased training volume, improved work capacity, and enhanced phosphocreatine availability.

Although a small number of interventions (n = 4) assessed creatine without training, a significant improvement was still observed (WMD = 12.36 kg). While this raises the possibility that creatine may influence muscle energetics or hydration status independent of training stimuli, these findings should be interpreted with caution given the small number of interventions (n = 4) and the lack of control over participants’ exercise habits outside the study protocols. The physiological explanations here remain speculative and cannot be confirmed from the current dataset. Nonetheless, it raises intriguing possibilities for creatine use in clinical or rehabilitation settings where physical activity may be limited.

Collectively, these findings highlight that creatine supplementation significantly improves squat strength, especially in younger, male, and trained individuals. Effects are magnified by higher dosing strategies and frequent training regimens. These results provide valuable insights for athletes, coaches, and clinicians seeking to enhance lower-body strength through evidence-based supplementation strategies.

### 4.5. Power-Based Performance

In our meta-analysis, creatine supplementation, especially when paired with exercise training, significantly enhanced power-based performance outcomes such as vertical jump and anaerobic output assessed via the Wingate test, particularly in younger adults. These improvements align with creatine’s known role in increasing phosphocreatine availability, which facilitates rapid ATP resynthesis during high-power, short-duration efforts like jumping and sprinting.

Subgroup analyses across both vertical jump and Wingate performance consistently revealed significant benefits for males but not for females. In the context of our findings, this discrepancy may be partly explained by physiological and hormonal differences, such as higher muscle mass, greater type II fiber composition, and more pronounced neuromuscular activation patterns in males along with possible methodological limitations in the included studies. Notably, the limited number of female-only studies (vertical jump: *n* = 5; Wingate: *n* = 3) hampers definitive conclusions and underscores the need for more well-powered trials in female cohorts. Additionally, females may exhibit higher baseline intramuscular creatine levels, potentially attenuating supplementation effects [110,111].

Interestingly, participants with prior training experience consistently showed greater improvements than untrained individuals in both power-based outcomes. This may be due to superior movement economy, neuromuscular efficiency, and higher fast-twitch fiber content in trained individuals, which may interact synergistically with creatine to amplify power output [18]. Conversely, untrained individuals may lack the neuromuscular adaptations necessary to fully exploit creatine’s benefits in these tasks.

In our dataset, a dose–response pattern emerged in both performance measures, with higher daily creatine intakes (>8 g/day) associated with significantly greater improvements in power-related activities than lower maintenance doses (≤8 g/day). This supports the notion that individuals with larger muscle mass or lower initial creatine stores may require higher dosages to achieve optimal muscle saturation and functional benefits.

Intervention duration also influenced both vertical jump and cycling peak power outcomes. For vertical jump, longer supplementation periods (≥8 weeks) yielded more favorable results than shorter ones, possibly due to cumulative neuromuscular adaptations. In contrast, all included Wingate studies were short-term (<8 weeks), and while favorable, the long-term effects of creatine supplementation on cycling anaerobic power as measured by the Wingate test remain unexamined.

Another unexpected yet consistent finding was that studies lacking structured training interventions reported more pronounced benefits than those incorporating training, which may reflect methodological differences, baseline participant characteristics, or limited reporting rather than a true physiological advantage. This may reflect methodological heterogeneity, differences in participant baseline fitness, or ceiling effects in trained individuals who have already adapted to high-intensity exercise. However, given the relatively small number of studies in the “no training” subgroup, this pattern could also be a chance finding, and the apparent effect should be viewed as preliminary until confirmed by larger, well-controlled trials.

Lastly, limitations in reporting across studies, particularly regarding training frequency, loading strategies, and program adherence restricted the ability to draw robust subgroup conclusions. Future research should aim for greater methodological transparency and include diverse populations, especially older adults, to explore whether creatine can offset age-related declines in power output using either or both of these ubiquitous and repeatable measures of human performance.

### 4.6. Limitations

This meta-analysis provides compelling evidence regarding the ergogenic effects of creatine supplementation on anaerobic performance. However, several limitations should be considered when interpreting the findings.

#### 4.6.1. Population Related Limitations

First, most studies focused on younger adults, with few including older individuals, limiting the generalizability of our findings to aging populations.

Second, most trials recruited male participants, while relatively few included females. Consequently, subgroup analyses in females were underpowered, which may have contributed to the lack of statistically significant effects observed in some female-specific outcomes despite trends toward improvement.

#### 4.6.2. Methodological Limitations

Third, many studies lacked detailed reporting on creatine dosage, loading strategies, and training frequency, limiting our ability to conduct nuanced subgroup analyses and assess dose–response effects.

Fourth, most interventions were short term, particularly for outcomes like Wingate and vertical jump tests, leaving the long-term effectiveness of creatine largely unresolved.

Fifth, an unexpected finding was greater Wingate performance improvements in studies without structured training compared to those with training. This counterintuitive result likely reflects differences in study design, baseline fitness levels, or other uncontrolled variables rather than true physiological effects.

Sixth, substantial variability existed in both the training protocols (e.g., exercise selection, intensity, and frequency) and creatine supplementation regimens (e.g., duration, form, and dose), which may have influenced the magnitude and consistency of observed effects. While we attempted to account for these differences through subgroup analyses, the lack of standardized intervention protocols may have limited the precision and generalizability of our conclusions.

Seventh, significant heterogeneity was observed across several analyses. This likely reflects variability in participant characteristics, training programs, and outcome assessments. Although subgroup analyses addressed some of this variability, residual heterogeneity remained.

Eighth, adherence to supplementation and dietary control were poorly reported, introducing variability and limiting the ability to isolate creatine’s independent effects. Furthermore, the type of creatine used and the dosing strategies were not consistently reported across studies. While creatine monohydrate was most commonly used, variations in formulation (e.g., buffered or ethyl ester forms) and inconsistent use of loading protocols or dose standardization may have introduced additional variability, limiting our ability to directly compare the efficacy of different creatine regimens.

### 4.7. Conclusion and Practical Applications

This systematic review and meta-analysis provides strong evidence that creatine supplementation enhances anaerobic performance, particularly in certain measures of maximal power output such as the Wingate test in specific subgroups and muscle strength such as squat and bench or chest press performance. The most pronounced effects were observed in younger trained males, and some outcomes such as squat strength and Wingate peak power showed larger gains with higher maintenance doses of creatine, especially in short-term interventions. However, benefits were not uniform across all outcomes or subgroups, and effects for measures like handgrip strength and leg press were small or non-significant in the pooled analysis. The generalizability of these findings remains limited due to the underrepresentation of females, older adults, and long-term interventions in the current literature. These gaps highlight the need for caution when extrapolating results to broader or understudied populations. While some outcomes demonstrated notable heterogeneity, effect directions were generally consistent and sensitivity analyses confirmed the robustness of the main findings.

From a practical standpoint, the observed improvements can translate into meaningful real-world benefits. For example, a 5 kg increase in squat strength may represent a 5 to 10 percent improvement in trained individuals, potentially enhancing athletic performance, reducing injury risk, and accelerating strength progression. In contrast, a 1.4 kg increase in bench press may be modest for highly trained populations but could still be meaningful for beginners, recreational lifters, or in rehabilitation settings.

For clinicians, creatine may be a useful adjunct for maintaining or improving muscle performance in patients undergoing rehabilitation or experiencing age-related muscle loss. While most included studies involved active populations performing resistance training, creatine’s well-documented safety profile and potential muscle-preserving effects support its consideration in clinical or rehabilitative settings, pending more targeted trials particularly in non-training contexts.

For athletes, creatine remains one of the most well-supported ergogenic aids, particularly in strength and power sports. Coaches and sport nutritionists can confidently recommend creatine, especially for those competing in short-duration high-intensity disciplines, with the caveat that effects may vary depending on exercise type, training status, and individual responsiveness.

For the general population, creatine supplementation may assist in improving strength, maintaining lean mass, and enhancing functional performance during physical activity. However, recommendations should be tailored to individual characteristics such as age, sex, habitual diet, and type of training. Overall, creatine supplementation is an effective and practical strategy for enhancing physical performance, but optimal use requires personalized implementation and further population-specific evidence.

### 4.8. Future Research Directions

Future research should prioritize including underrepresented groups, particularly older adults and females. Well-powered, sex-specific trials are needed to determine if creatine’s effects differ by sex and whether hormonal or physiological factors influence outcomes. Long-term (≥8 weeks), large-scale interventions, especially those with periodized or progressive training are needed to assess the sustainability and magnitude of performance gains over time. Future studies should also explore the effects of creatine across diverse training modalities, such as aerobic, skill-based, or concurrent exercise since most current evidence is limited to resistance-based protocols, restricting generalizability. Finally, standardizing supplementation protocols, outcome measures, and training variables will improve comparability, reduce heterogeneity, and enhance the applicability of future findings.

## Figures and Tables

**Figure 1 nutrients-17-02748-f001:**
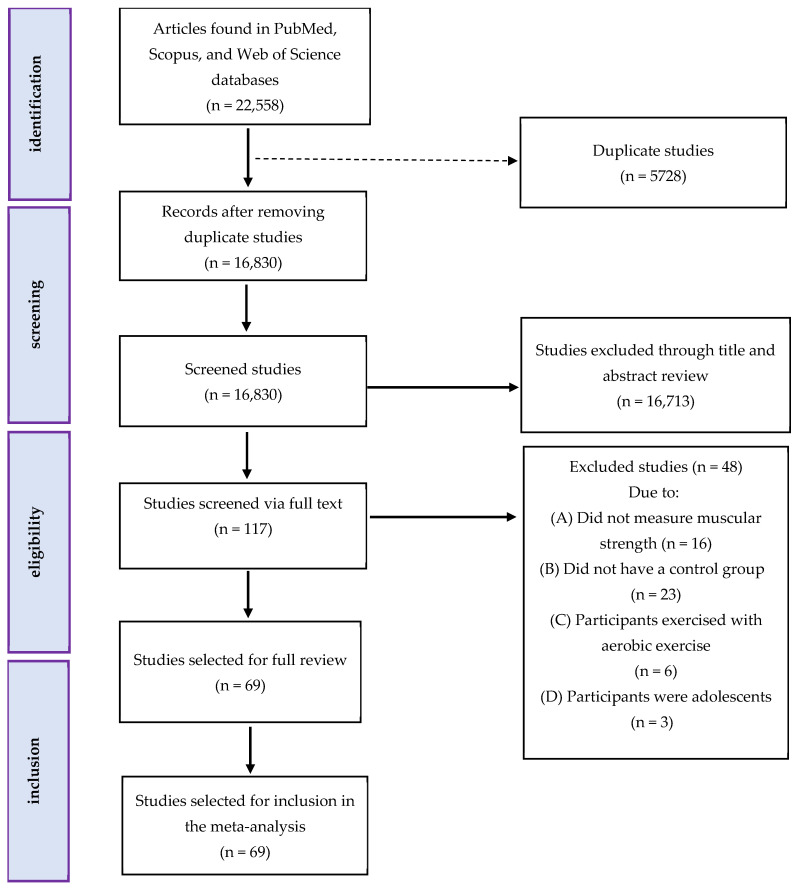
Flow diagram of systematic literature search.

**Figure 2 nutrients-17-02748-f002:**
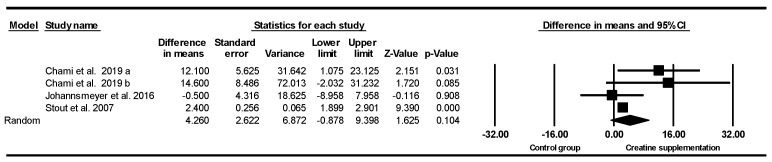
Forest plot of effects of creatine supplementation plus exercise vs. control group on handgrip strength [85,86,91]. Data are reported as WMD (95% confidence limits). WMD, weighted mean differences.

**Figure 3 nutrients-17-02748-f003:**
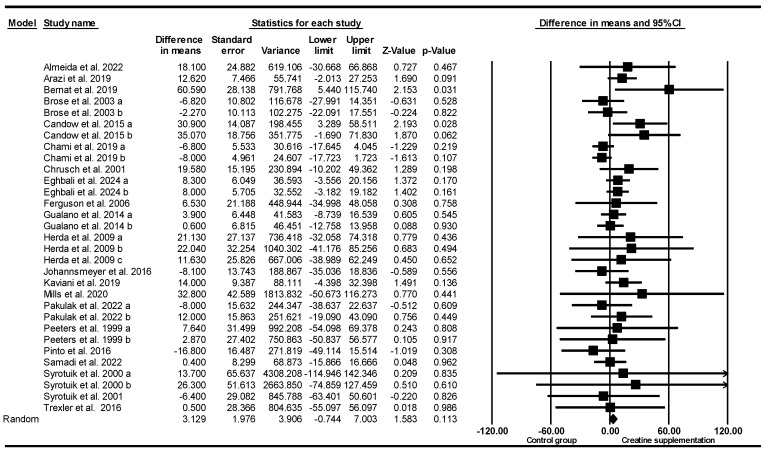
Forest plot of effects of creatine supplementation plus exercise vs. control group on leg press strength [6,30,31,34,38,41,44,47,55,56,60,64,77,83,84,85,86,88,89,90,92]. Data are reported as WMD (95% confidence limits). WMD, weighted mean differences.

**Figure 4 nutrients-17-02748-f004:**
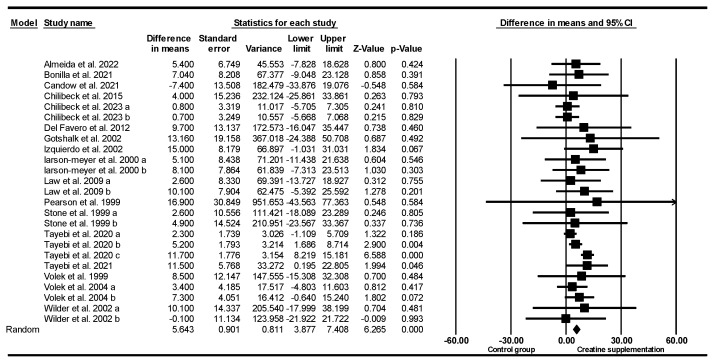
Forest plot of effects of creatine supplementation plus exercise vs. control group on squat. [29,30,35,37,40,42,45,49,54,58,61,62,66,67,70,76,80]. Data are reported as WMD (95% confidence limits). WMD, weighted mean differences.

**Figure 5 nutrients-17-02748-f005:**
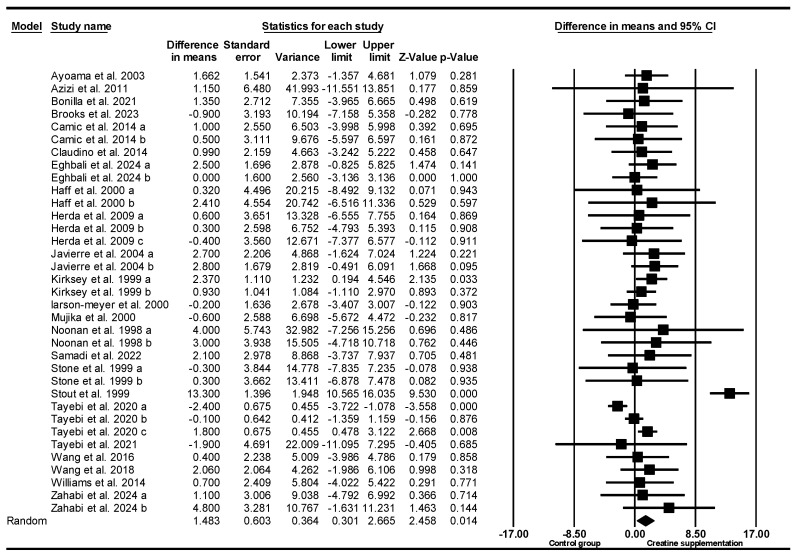
Forest plot of effects of creatine supplementation plus exercise vs. control group on vertical jump [18,32,35,36,39,41,44,46,51,52,56,58,59,61,62,68,69,71,73,75,80,81,87]. Data are reported as WMD (95% confidence limits). WMD, weighted mean differences.

**Figure 6 nutrients-17-02748-f006:**
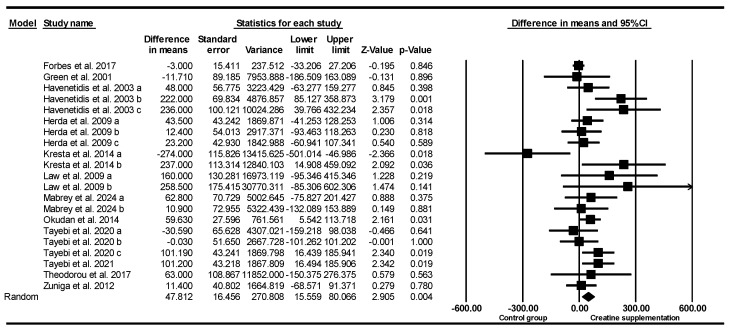
Forest plot of effects of creatine supplementation plus exercise vs. control group on Wingate peak power [16,19,43,44,49,50,53,61,62,63,78,79]. Data are reported as WMD (95% confidence limits). WMD, weighted mean differences.

**Table 1 nutrients-17-02748-t001:** Study, participant, and intervention characteristics.

Source, Year	Study Characteristics				Participant Characteristics			Creatine Type and Maintenance Dose	Loading Dose	Exercise Characteristics	Energy Intake (kcal/d)
	Sample Size (Biological Sex)	Groups	Intervention Duration	Outcomes	Health Status	Age (Years) Mean ± SD	BMI (kg/m^2^) Mean ± SD				
Almeida et al., 2022 [30]	34 M	Creatine CON	4 weeks	Squat Leg press Bench press	Healthy (trained)	Creatine: 23.1 ± 3.1 CON: 23.8 ± 3.3	Creatine: NR CON: NR	Creatine monohydrate: 0.03–0.3 g/(kg·day)	0.3 g/(kg·day) (7 days) 0.03 g/(kg·day) (21 days)	RT: 6 exercises × 3 sets of 8–12 reps at 85% of 1RM × 3 d/w	NR
Amiri et al., 2023 [82]	30 M and F	Creatine CON	10 weeks	Bench press	Healthy (untrained)	Creatine: 68.1 ± 7.2 CON: 68.1 ± 7.2	Creatine: 27.8 ± 5.1 CON: 28.3 ± 2.9	Creatine monohydrate: 0.1 g/(kg·day)	No loading	RT: 7 exercises × 3 sets of 10 reps at 75% of 1RM and rest intervals between sets and between movements were 2 and 3 min, respectively × 3 d/w	NR
Arazi et al., 2019 [31]	16 M	Creatine CON	6 weeks	Leg press Bench press	Healthy (untrained)	Creatine: 20.8 ± 2.1 CON: 20.3 ± 3	Creatine: <18.5 CON: <18.5	Creatine ethyl ester: 5 g/day	20 g/day (5 days)	RT: 6 exercises × 3 sets of 8–10 reps at 60–80%1RM × 3 d/w	NR
Ayoama et al., 2003 [32]	21 M	Creatine CON	6 days	Vertical Jump	Healthy (trained)	Creatine: 23.4 ± 4.7 CON: 22.8 ± 4.3	Creatine: NR CON: NR	Creatine: 20 g/day	No loading	Without exercise training	NR
Azizi et al., 2011 [73]	20 F	Creatine CON	6 days	Vertical Jump Bench Press	Healthy (trained)	Creatine: 20.9 ± 1.6 CON: 20.9 ± 1.6	Creatine: NR CON: NR	Creatine monohydrate: 5 g/day	No loading	Swimming: Participants were required to maintain their normal training and physical activity patterns.	NR
Becque et al., 2000 [33]	23 M	Creatine CON	6 weeks	Bench press	Health (trained)	Creatine: 21.5 ± 2.7 CON: 21.5 ± 2.7	Creatine: NR CON: NR	Creatine monohydrate: 2 g/day	20 g/day (5 days)	RT: Participants trained the arm with a standard wide-grip cambered lifting bar on a preacher curl bench.	NR
Bernat et al., 2019 [34]	24 M	Creatine CON	8 weeks	Leg press Bench press	Healthy (untrained)	Creatine: 59 ± 7.1 CON: 58.1 ± 5.8	Creatine: NR CON: NR	Creatine monohydrate: 0.1 g/(kg·day)	No loading	HVRT: 6 Exercises × 3–4 sets at 80% baseline 1RM to volitional fatigue and 2 min of rest separated each set × 2 d/w	Creatine 1991.3 ± 311.4 CON 2272 ± 557.2
Bonilla et al., 2021 [35]	16 M	Creatine CON	8 weeks	Squat Vertical jump	Healthy (trained)	Creatine: 26.1 ± 7.6 CON: 26.1 ± 7.6	Creatine: 23.8 ± 1.7 CON: 24.2 ± 2.5	Creatine monohydrate: 0.1 g/(kg·day)	No loading	cluster-set RT: 3 movements × 3 sets × 12 reps and clusters per set 4 × 3RM and intra-set rest 20 s and inter-set rest 180 s × 2 d/w	All participants were prescribed to consume ∼39 kcal·kg^−1^ FFM per day. 5.0 g·kg^−1^ FFM·day^−1^ of carbohydrates, 2.5 g·kg^−1^ FFM·day^−1^ of protein, and 1.0 g·kg^−1^ FFM·day^−1^ of fat
Brenner et al., 2000 [74]	16 F	Creatine CON	5 weeks	Bench press	Healthy (trained)	Creatine: 18.1 ± 1.7 CON: 19.5 ± 1.9	Creatine: NR CON: NR	Creatine monohydrate: 2 g/day	20 g/day (1 week)	RT: 4–5 sets × 5–10 reps at 40–85%1RM and rested for 3–5 min between sets × 3 d/w	NR
Brooks et al., 2023 [75]	13 F	Creatine CON	6 weeks	Vertical jump	Healthy (trained)	Creatine: 21 ± 1 CON: 20 ± 1	Creatine: 22.5 ± 2.7 CON: 22.3 ± 3.7	Creatine monohydrate: 0.1 g/(kg·day)	No loading	Collegiate dance	Calories (kcals) pre: creatine 1374.1 ± 508.8 and CON 1740.7 ± 1035.5
Brose et al., 2003 [83]	26 M and F	Creatine M CON M Creatine F CON F	14 weeks	Leg press Bench press	Healthy (untrained)	Creatine_M_: 68.7 ± 4.8 CON_M_: 68.3 ± 3.2 Creatine_F_: 70.8 ± 6.1 CON_F_: 69.9 ± 5.6	Creatine_M_: NR CON_M_: NR Creatine_F_: NR CON_F_: NR	Creatine monohydrate: 5 g/day	No loading	RT: 12 exercises were used to train the major muscle groups of the upper and lower body in a circuit set system, using weight training machines for 3 sets at 50–80% of the initial 1RM strength × 3 d/w.	Creatine M: 2249 ± 488 CON M: 2603 ± 587 Creatine F: 1821 ± 457 CON F: 1788 ± 507
Camic et al., 2014 [36]	77 M	Creatine1 Creatine2 CON	4 weeks	Vertical jump Bench press	Healthy (untrained)	Creatine_1_: 22.1 ± 2.5 Creatine_2_: 22.1 ± 2.5 CON: 22.1 ± 2.5	Creatine_1_: NR Creatine_2_: NR CON: NR	Polyethylene glycosylated creatine: 1.25–2.50 g/day	No loading	Without exercise training	The participants were encouraged to continue with their normal exercise habits.
Candow et al., 2015 [84]	39 M and F	Creatine1 Creatine2 CON	12 weeks	Leg press Bench press	Healthy (untrained)	Creatine_1_: 53.2 ± 2.5 creatine_2_: 55.5 ± 3.5 CON: 57.2 ± 6.5	Creatine_1_: 26.5 ± 4.2 creatine_2_: 29 ± 5.9 CON: 26.8 ± 3.4	Creatine monohydrate: 0.1 g/(kg·day)	No loading	RT: 11 exercises × 3 sets × 10 reps to muscle fatigue with 1–2 min rest between sets for each exercise at an intensity corresponding to their 10 reps maximum for each exercise × 3 d/w	Creatine1: 1912.6 ± 360.2 Creatine2: 1919.6 ± 390.1 CON: 1937.2 ± 576.7
Candow et al., 2021 [37]	38 M	Creatine CON	52 weeks	Squat Bench press	Healthy (untrained)	Creatine: 58 ± 6 CON: 56 ± 5	Creatine: NR CON: NR	Creatine monohydrate: 0.1 g/(kg·day)	No loading	RT: 3 sets × 10 reps at 80%1RM	Total energy (kcal) for Creatine group 2212 and for CON group 1859 at the baseline
Chami et al., 2019 [85]	33 M and F	Creatine1 Creatine2 CON	10 days	Leg press Bench press Handgrip strength	Healthy (untrained)	Creatine_1_: 59.3 ± 3.2 Creatine_2_: 58.8 ± 5.9 CON: 57.3 ± 4.6	Creatine_1_: 28.3 Creatine_2_: 28.2 CON: 28.8	Creatine monohydrate: Creatine_1_: 0.3 g/(kg·day) Creatine monohydrate _2_: 0.1 g/(kg·day)	No loading	Without exercise training	NR
Chilibeck et al., 2015 [76]	33 F	Creatine CON	52 weeks	Squat Bench press	Postmenopausal women (untrained)	Creatine: 57 ± 4 CON: 57 ± 7	Creatine: NR CON: NR	Creatine monohydrate: 0.14 g/(kg·day)	No loading	RT: 3 sets × 10 reps to muscle fatigue were performed at approximately 80%1RM, and intensity of exercises was increased progressively on an individual basis × 2 d/w.	Creatine: 1791 ± 440 CON: 1845 ± 231
Chilibeck et al., 2023 [29]	237 F	Creatine CON	52 weeks 104 weeks	Squat Bench press	Healthy (untrained)	Creatine: 59 ± 5.6 CON: 59 ± 5.7	Creatine: NR CON: NR	Creatine monohydrate: 0.14 g/(kg·day)	No loading	RT: exercises in 2 sets of 10 reps × 2 d/w 3 d of non-supervised brisk walking was performed outside of the laboratory	All participants received a supplement of 500 mg of calcium and 400 IU of vitamin D per day
Chrusch et al., 2001 [38]	30 M	Creatine CON	12 weeks	Leg press Bench press	Healthy (untrained)	Creatine: 70.4 ± 6.4 CON: 71.1 ± 6.7	Creatine: 27.9 ± 4.4 CON: 25.7 ± 2.6	Creatine monohydrate: 0.07 g/(kg·day)	0.3 g/(kg·day) (5 days)	RT: were performed at 50% of pre 1RM. Training volumes were progressed throughout the study at equal levels for all participants, and the resistance was individually progressed × 2 d/w.	Creatine: 2138.6 ± 145.6 CON: 2173.7 ± 123.1
Claudino et al., 2014 [39]	14 M	Creatine CON	7 weeks	Vertical jump	Healthy (trained)	Creatine: 18.3 ± 0.9 CON: 18.3 ± 0.9	Creatine: NR CON: NR	Creatine monohydrate: 5 g/day	20 g/day (1 week)	RT: performed between 50 and 60 min and involved multiple joint exercises with 3 sets × 8–10 reps maximum interspersed by 1 to 3 min of recovery. Additionally, plyometric exercises were performed × 2 d/w. The specific training consisted of small-sided games performed × 4–5 d/w.	Creatine: 2718.4 ± 603.2 CON: 2887.9 ± 700.6
Del Favero et al., 2012 [40]	17 M	Creatine CON	10 days	Squat Bench press	Healthy (untrained)	Creatine: 18–30 CON: 18–30	Creatine: NR CON: NR	Creatine monohydrate: 20 g/day	No loading	Without exercise training	Creatine: 3.170 ± 441 CON: 2.780 ± 708
Earnest et al., 1995 [28]	8 M	Creatine CON	2 weeks	Bench press	Health (trained)	Creatine: 29.5 ± 3.6 CON: 31.8 ± 2.2	Creatine: NR CON: NR	Creatine monohydrate: 20 d/day	No loading	Without exercise training	NR
Eghbali et al., 2024 [41]	36 M	Creatine hydrochloride Creatine monohydrate CON	8 weeks	Leg press Vertical jump Bench press	Healthy (trained)	Creatine_1_: 25.3 ± 2.4 Creatine_2_: 23.8 ± 3.6 CON: 24.1 ± 4.1	Creatine_1_: 22.16 ± 1.5 Creatine_2_: 22.9 ± 1.3 CON: 23.1 ± 1.3	Creatine hydrochloride or Creatine monohydrate: 0.03 g/(kg·day)	No loading	RT: 9 exercises × 3 sets of 6–12 reps at 70–85%1RM and 2–3 min rest between sets × 3 d/w	Calories (Kcal) pre: Creatine 1: 2625 ± 119.02 Creatine 2: 2730.5 ± 104.3 CON: 2755.6 ± 96.5
Ferguson et al., 2006 [77]	26 F	Creatine CON	10 weeks	Leg press Bench press	Healthy (trained)	Creatine: 24.6 ± 3.4 CON: 24.6 ± 3.4	Creatine: NR CON: NR	Creatine monohydrate: 0.03 g/(kg·day)	0.3 g/(kg·day) (1 weeks)	RT: 12-exercise × 4 d/w	Creatine: 1721.8 ± 252 CON: 1904.2 ± 248.9
Forbes et al., 2017 [78]	17 F	Creatine CON	4 weeks	Wingate	Healthy (untrained)	Creatine: 23.0 ± 4.0 CON: 23.0 ± 4.0	Creatine: 23.4 ± 2.4 CON: 23.4 ± 2.4	Creatine monohydrate: 5 g/day	20 g/day (5 days)	HIIT: sessions of the week included repeated 30 s Wingate, repeated sprints, and 60 s off repeats. Participants completed repeated 30 s all-out cycling “sprint” intervals with 4-min recovery intervals × 4 d/w	Creatine: 1912 ± 424 CON: 1978 ± 571
Furtado et al., 2024 [72]	12 M	Creatine CON	5 days	Bench press	Healthy (trained)	Creatine: 25.2 ± 3.4 CON: 25.2 ± 3.4	Creatine: NR CON: NR	Creatine monohydrate: 20 g/day	No loading	RT: They were instructed not to change their habitual diet or physical activity routine.	Participants were asked for no change in their dietary habits throughout the study.
Gotshalk et al., 2002 [42]	18 M	Creatine CON	7 days	Squat Bench press	Healthy (untrained)	Creatine: 65.4 ± 4.7 CON: 65.7 ± 5.6	Creatine: NR CON: NR	Creatine monohydrate: 0.3 g/(kg·day)	No loading	Without exercise training	NR
Green et al., 2001 [16]	19 M	Creatine CON	6 days	Wingate	Healthy (untrained)	Creatine: 26.3 ± 5.3 CON: 24.1 ± 3.1	Creatine: NR CON: NR	Creatine monohydrate: 20 g/day	No loading	Without exercise training	NR
Gualano et al., 2014 [6]	60 F	Creatine1 CON1 Creatine2 CON2	24 weeks	Leg press Bench press	Postmenopausal women (untrained)	Creatine_1_: 66.1 ± 4.8 CON_1_: 66.3 ± 6 Creatine_2_: 67.1 ± 5.6 CON_2_: 63.6 ± 3.6	Creatine_1_: 27.1 ± 3.5 CON_1_: 26.8 ± 5.5 Creatine_2_: 28 ± 2.1 CON_2_: 28.2 ± 3.6	Creatine monohydrate: 5 g/day	20 g/day (5 days)	RT: 7 exercises × 3 sets of 8–12 RM, except during the first week, when a reduced volume of two sets of 15–20 RM for each exercise was performed	Creatine1: 1462 ± 246 CON1: 1438 ± 343 Creatine2: 1257 ± 318 CON2: 1461 ± 202
Haff et al., 2000 [18]	36 M and F	Creatine CON	6 weeks	Vertical jump	Healthy (untrained)	Creatine: 19.9 ± 0.4 CON: 19.9 ± 0.4	Creatine: NR CON: NR	Creatine monohydrate: 0.3 g/(kg·day)	No loading	RT and ST × 5 d/w ST: 6 × 100–150 m with 90–120 s rest	Creatine: 2682 ± 233 CON:2737 ± 238
Havenetidis et al., 2003 [43]	21 M	Creatine (10 g) Creatine (25 g) Creatine (35 g) CON (10 g) CON (25 g) CON (35 g)	4 days	Wingate	Health (trained)	Creatine: 29.4 ± 3.7 CON: 29.4 ± 3.7	Creatine: NR CON: NR	Creatine monohydrate 10 g/day or 25 g/day or 35 g/day	No loading	Without exercise training	NR
Herda et al., 2009 [44]	58 M	Creatine1 Creatine2 Creatine3 CON	30 days	Leg press Vertical jump Wingate Bench press	Healthy (untrained)	Creatine_1_: 21 ± 2 Creatine_2_: 21 ± 2 Creatine_3_: 21 ± 2 CON: 21 ± 2	Creatine_1_: NR Creatine_2_: NR Creatine_3_: NR CON: NR	Creatine_1_ ( creatine monohydrate): 5 g/day Creatine_2_ ( PEG creatine hydrochloride): 1.25 g/day Creatine3: PEG creatine hydrochloride 2.5 g/day	No loading	Without exercise training	NR
Izquierdo et al., 2002 [45]	19 M	Creatine CON	5 days	Squat Bench press	Healthy (trained)	Creatine: 20.8 ± 5 CON: 23.6 ± 5	Creatine: NR CON: NR	Creatine monohydrate: 20 g/day	No loading	Handball: Four times a week RT: once a week for strength and endurance training and played in one official handball game per week	NR
Javierre et al., 2004 [46]	19 M	Creatine CON	5 days	Vertical jump	Healthy (trained)	Creatine: 22.9 ± 3.1 CON: 22.9 ± 3.1	Creatine: NR CON: NR	Creatine monohydrate: 20 g/day	No loading	Without exercise training	NR
Johannsmeyer et al., 2016 [86]	29 M and F	Creatine CON	12 weeks	Leg press Bench press Handgrip strength	Healthy (untrained)	Creatine: 58 ± 3 CON: 57.6 ± 5	Creatine: NR CON: NR	Creatine monohydrate: 0.1 g/(kg·day)	No loading	RT: 4 exercises × 2 sets of drop-set RT for each exercise. Sets: reps with 80% baseline 1RM no rest and reps with 30% 1RM after rest (1–2 min) or no rest	Creatine: 2097.5 ± 329.5 CON: 1906.9 ± 732.8
Kaviani et al., 2019 [47]	18 M	Creatine CON	8 weeks	Leg press Bench press	Healthy (untrained)	Creatine: 23 ± 3 CON: 23 ± 3	Creatine: NR CON: NR	Creatine monohydrate: 0.07 g/(kg·day)	No loading	RT: 6 exercise trainings consisted of 3 sets of 10 reps at 75% 1RM × 3 d/w	NR
Kelly et al., 1998 [48]	18 M	Creatine CON	4 weeks	Bench press	Healthy (trained)	Creatine: 26.8 ± 6.1 CON: 26.8 ± 6.1	Creatine: NR CON: NR	Creatine monohydrate: 5 g/day	20 g/day (5 days)	RT: 3 sets × 6–10 reps of 85%1RM	NR
Kirksey et al., 1999 [87]	36 M and F	Creatine CON	6 weeks	Vertical jump	Healthy (trained)	Creatine: 19.9 ± 0.4 CON: 19.9 ± 0.4	Creatine: NR CON: NR	Creatine monohydrate: 0.3 g/(kg·day)	No loading	Without exercise training	Creatine: 2838 ± 260 CON: 2699 ± 206
Kresta et al., 2014 [79]	15 F	Creatine + Beta alanine Beta alanine Creatine CON (+BetaAlanine)	4 weeks	Wingate	Health (untrained)	Creatine + Beta alanine: 18–35 Beta alanine: 18–35 Creatine: 18–35 CON: 18–35	Creatine + Beta alanine: NR Beta alanine: NR Creatine: NR CON: NR	Creatine monohydrate: 0.1 g/(kg·day)	0.3 g/(kg·day) (1 week)	Without exercise training	NR
Larson-meyer et al., 2000 [80]	14 F	Creatine CON	5 weeks 13 weeks	Vertical jump Squat Bench press	Healthy (trained)	Creatine: 19.3 ± 1.4 CON: 19.0 ± 1.5	Creatine: NR CON: NR	Creatine monohydrate: 5 g/day	15 g/day (7 days)	RT: 1–2 sets × 6–20 reps of 50–85%1RM × 2–3 d/w Soccer: 110 min × 0–3 d/w	NR
Law et al., 2009 [49]	17 M	Creatine CON	2 days 5 days	Wingate Squat Bench press	Healthy (trained)	Creatine: 23.1 ± 3.5 CON: 26.4 ± 3.9	Creatine: NR CON: NR	Creatine: 20 g/day	No loading	RT: 9 exercises × 3 sets × 12–15 reps and 1–2 min rest between sets × 2 days	NR
Mabrey et al., 2024 [50]	12 M	Creatine and caffeine Caffeine Creatine CON (+caffeine)	7 days	Wingate	Healthy (trained)	Creatine and caffeine: 21.9 ± 0.7 caffeine: 21.9 ± 0.7 Creatine: 21.9 ± 0.7 CON: 21.9 ± 0.7	Creatine and caffeine: 26.6 ± 5.1 caffeine: 26.6 ± 5.1 Creatine: 26.6 ± 5.1 CON: 26.6 ± 5.1	Creatine nitrate: 5 g/day	No loading	RT: Ten reps at 50% of the predicted 1RM, followed by five reps at 70%, and one rep at 90%.	Energy Intake (kcal/d/kg): Creatine and caffeine: 20.4 ± 5.1 Caffeine: 23.2 ± 8.2 Creatine: 21.9 ± 7.2 CON: 24.1 ± 9.1
Mills et al., 2020 [88]	22 M and F	Creatine CON	6 weeks	Leg press Bench press	Healthy (trained)	Creatine: 26.1 ± 4.6 CON: 26.4 ± 5.1	Creatine: NR CON: NR	Creatine monohydrate: 0.1 g/(kg·day)	No loading	RT: Split training of 6 movement × 3 sets to volitional fatigue per exercise and 2 min rest between sets × 5 d/w	NR
Mujika et al., 2000 [51]	17 M	Creatine CON	2 weeks	Vertical jump	Healthy (trained)	Creatine: 20.3 ± 1.4 CON: 20.3 ± 1.4	Creatine: NR CON: NR	Creatine monohydrate: 20 g/day	No loading	Without exercise training	NR
Noonan et al., 1998 [52]	39 M	Low Creatine High Creatine CON	8 weeks	Vertical jump Bench press	Healthy (trained)	Low Creatine: 19.4 ± 1.02 High Creatine: 19.7 ± 1.4 CON: 20.4 ± 1.3	Low Creatine: NR High Creatine: NR CON: NR	Low creatine monohydrate: 0.1 g/(kg·day) High creatine monohydrate: 0.3 g/(kg·day)	20 g/day (5 days)	RT: 5 exercises × 3 sets × 2–10 reps and 1.5–3 min rest between sets × 2 d/w Sprints and agility: 20–30 min × 2 d/w	NR
Okudan et al., 2014 [53]	22 M	Creatine CON	4 weeks	Wingate	Health (untrained)	Creatine: 22 ± 2.2 CON: 21.5 ± 2	Creatine: 22.3 ± 2.3 CON: 23.5 ± 2.7	Creatine: 5 g/day	20 g/day (6 days)	Without exercise training	NR
Pakulak et al., 2022 [89]	33 M and F	Creatine and caffeine Caffeine Creatine CON (+caffeine)	6 weeks	Leg press Bench press	Healthy (trained)	Creatine and caffeine: 22 ± 4 Caffeine: 19 ± 1 Creatine: 22 ± 4 CON: 23 ± 7	Creatine and caffeine: NR Caffeine: NR Creatine: NR CON: NR	Creatine monohydrate: 0.1 g/(kg·day)	No loading	RT: Split days involved 6 exercises and 3 sets of 6–10 reps to volitional fatigue per exercise. × 4 d/w	Participants used a 3-day food booklet to record food intake for two weekdays and one weekend day.
Pearson et al., 1999 [54]	16 M	Creatine CON	10 weeks	Squat Bench press	Healthy (trained)	Creatine: 20.7 CON: 20.7	Creatine: NR CON: NR	Creatine monohydrate 5 g/day	20 g/day (5 days)	RT: 5–12 exercises × 3–4 sets × 2–12 reps at 60–89%1RM × 4 d/w	NR
Peeters et al., 1999 [55]	35 M	Creatine CON	6 weeks	Leg press Bench press	Healthy (trained)	Creatine: 21.2 ± 2.6 CON: 21.2 ± 2.6	Creatine: NR CON: NR	Creatine monohydrate: 10 g/day Creatine: phosphate 10 g/day	20 g/day (3 days)	RT: 3 movements × 2–10 reps× 5 sets at 50–90%1RM × 5 d/w	NR
Pinto et al., 2016 [90]	27 M and F	Creatine CON	12 weeks	Leg press Bench press	Healthy (untrained)	Creatine: 67.4 ± 4.7 CON: 67.1 ± 6.3	Creatine: NR CON: NR	Creatine monohydrate: 5 g/day	No loading	RT: The duration of one training session was 60 min, and they performed 3 sets of 10–15 RM with 60 s of rest between the sets. The participants underwent training either for the upper limbs and abdomen or lumbar region. ×3 d/w	Creatine: 1589.6 ± 395.2 CON: 1472.3 ± 531.8
Samadi et al., 2022 [56]	20 M	Creatine CON (+BetaAlanine)	1 weeks	Leg Press Vertical jump Bench press	Healthy (trained)	Creatine: 21.4 ± 2.1 CON: 21.6 ± 2	Creatine: 23.8 ± 1.2 CON: 23.5 ± 2	Creatine monohydrate: 0.3 g/(kg·day)	No loading	Without exercise training	NR
Selsby et al., 2004 [57]	20 M	Creatine CON	10 days	Bench press	Healthy (trained)	Creatine: 20.6 ± 2.3 CON: 20.2 ± 1.5	Creatine: NR CON: NR	Creatine monohydrate: 2.5 g/day	No loading	Without exercise training	NR
Stone et al., 1999 [58]	42 M	Creatine+ Calcium pyruvate Calcium pyruvate Creatine CON (+Calcium pyruvate)	5 weeks	Squat Vertical jump Bench press	Healthy (trained)	Creatine+ Calcium pyruvate: 18.3 ± 0.5 Calcium pyruvate: 18.6 ± 0.6 Creatine: 18.7 ± 1.0 CON: 18.3 ± 0.4	Creatine+ Calcium pyruvate: NR Calcium pyruvate: NR Creatine: NR CON: NR	Creatine monohydrate: 0.09 g/(kg·day)	No loading	RT and power training: 5–7 exercises × 3 sets × 1–10 reps at 60–89%1RM × 3 d/w Football training: × 2–3 d/w	NR
Stout et al., 1999 [59]	16 M	Creatine CON	8 weeks	Vertical jump Bench press	Healthy (trained)	Creatine: 19.6 ± 1.0 CON: 19.6 ± 1.0	Creatine: NR CON: NR	Creatine monohydrate: 10.5 g/day	21 g/day (5 days)	RT: split exercises × 3 sets × 2–10 reps with 1.5–3 min rest between sets × 4 d/w running and agility training: 20–30 min × 2 d/w	
Stout et al., 2007 [91]	15 M and F	Creatine CON	2 weeks	Handgrip strength	Healthy (untrained)	Creatine: 74.5 ± 6.4 CON: 74.5 ± 6.4	Creatine: NR CON: NR	Creatine citrate: 10 g/day	20 g/day (7 days)	Without exercise training	Participants were asked to maintain their normal dietary pattern.
Syrotuik et al., 2000 [60]	21 M	Loading creatine Maintenance creatine CON	5 weeks	Leg press Bench press	Healthy (trained)	Loading creatine: 22.7 ± 0.2 Maintenance creatine: 22.7 ± 0.5 CON: 23.6 ± 1.1	Loading creatine: NR Maintenance creatine: NR CON: NR	Creatine monohydrate Maintenance group: 0.03 g/(kg·day)	Loading and maintenance group: 0.3 g/(kg·day) (5 days)	RT: 6–8 exercises × 2–5 sets × 2–12 reps at 65–95%1RM × 4 d/w	NR
Syrotuik et al., 2001 [92]	22 M and F	Creatine CON	6 weeks	Leg press Bench press	Healthy (trained)	Creatine: 23 CON: 23	Creatine: NR CON: NR	Creatine monohydrate: 0.03 g/kg/day	0.3 g/(kg·day) (5 days)	RT: 9 movements × 2 d/w Interval rowing 4 d/w	NR
Tayebi et al., 2020 [61]	36 M	High creatine monohydrate Low creatine monohydrate creatine hydrochloride CON	7 days	Vertical jump Squat Wingate Bench press	Healthy (trained)	High creatine monohydrate: 22.1 ± 2.1 Low creatine monohydrate: 22.1 ± 2.1 Creatine hydrochloride: 22.1 ± 2.1	High creatine monohydrate: 22.3 ± 3.4 Low creatine monohydrate: 22.3± 3.4 Creatine hydrochloride: 22.3 ± 3.4	High creatine monohydrate: 20 g/day Low creatine monohydrate: 3 g/day Creatine hydrochloride: 3 g/day	No loading	RT	NR
Tayebi et al., 2021 [62]	18 M	Creatine CON	2 weeks	Squat Vertical jump Wingate Bench press	Healthy (trained)	Creatine: 21.5 ± 1.1 CON: 22.3 ± 2.1	Creatine: NR CON: NR	Creatine hydrochloride: 3 g/day	No loading	Without exercise training	NR
Theodorou et al., 2017 [63]	14 M	Creatine CON	4 days	Wingate	Healthy (trained)	Creatine: 24 ± 5 CON: 24 ± 5	Creatine: NR CON: NR	Creatine monohydrate: 25 g/day	No loading	Without exercise training	NR
Trexler et al., 2016 [64]	28 M	Creatine CON	5 days	Leg press Bench press	Healthy (trained)	Creatine: 20.3 ± 2.3 CON: 19.9 ± 2	Creatine: NR CON: NR	Creatine monohydrate 20 g/day	No loading	Without exercise training	Creatine 2518.9 ± 372.7 CON 2891.9 ± 941.9
Vilar Neto et al., 2018 [65]	36 M	Creatine1 Creatine2 CON	6 weeks	Bench press	Healthy (trained)	Creatine1: 24.4 ± 6.1 Creatine2: 21.4 ± 2.8 CON: 21.6 ± 3	Creatine1: 25.7 ± 3 Creatine2: 24 ± 2.4 CON: 24.2 ± 1.5	Creatine monohydrate: 3 g/day or 5 g/day	No loading	RT: the participants performed 4 sets of 8 to 10 reps. The participants were also instructed to select a load that would enable them to perform a minimum of 7 reps; when the reps exceed 10, the load was increased.	NR
Volek et al., 1999 [66]	19 M	Creatine CON	12 weeks	Squat Bench press	Healthy (trained)	Creatine: 25.6 ± 4.8 CON: 25.4 ± 5.9	Creatine: NR CON: NR	Creatine monohydrate: 5 g/day	25 g/day (1 week)	RT: × 3–4 d/w	Creatine: 3156 CON: 3156
Volek et al., 2004 [67]	17 M	Creatine CON	4 weeks	Squat Bench press	Healthy (trained)	Creatine: 20.7 ± 1.9 CON: 21.3 ± 3	Creatine: NR CON: NR	Creatine monohydrate: 0.05 g/(kg·day)	0.3 g/(kg·day) (1 week)	RT: 5 exercises × 3 sets × 8–10 reps with 1–3 min rest between sets × 2–4 d/w	NR
Wang et al., 2016 [69]	30 M	Creatine CON	6 days	Vertical jump	Healthy (trained)	Creatine 19.9 ± 1.8 CON: 19.4 ± 1.1	Creatine: NR CON: NR	Creatine monohydrate: 2 g/day	20 g/day (6 days)	Without exercise training	NR
Wang et al., 2018 [68]	30 M	Creatine CON	4 weeks	Vertical jump	Healthy (trained)	Creatine: 20 ± 2 CON: 20 ± 1	Creatine: NR CON: NR	Creatine monohydrate: 2 g/day	20 g/day (6 days)	RT: 6 sets of 5-RM half squats and plyometric jumps × 3 d/w	NR
Wilder et al., 2002 [70]	27 M	Low-dose creatine High-dose creatine CON	10 weeks	Squat	Healthy (trained)	Creatine_1_: 18.8 ± 1 Creatine_2_: 19 ± 1.1 CON: 19.2 ± 1.1	Creatine_1_: NR Creatine_2_: NR CON: NR	Creatine: 5 or 3 g/day	20 g/day (6 days)	RT: 18 movements × 1–8 sets × 1–10 reps at 75–90%1RM × 4 d/w	NR
Williams et al., 2014 [71]	16 M	Creatine CON	7 days	Vertical jump	Healthy (trained)	Creatine 25.4 ± 4.5 CON 26.7 ± 4.6	Creatine: NR CON: NR	Creatine monohydrate: 20 g/day	No loading	Without exercise training	NR
Zahabi et al., 2024 [81]	12 F	Creatine CON	3 weeks 6 weeks	Vertical jump Bench press	Healthy (trained)	Creatine: 18.8 ± 0.7 CON: 18.9 ± 0.6	Creatine: 21.4 ± 1.3 CON: 21.8 ± 3.6	Creatine monohydrate: 10 g/training days	20 g/day (5 days)	RT: 5 exercises × 3 sets of 12–14 reps at 65–75%1RM and 30–60 s rest between sets × 4 d/w	NR
Zuniga et al., 2012 [19]	22 M	Creatine CON	7 days	Wingate Bench press	Healthy (untrained)	Creatine: 22.1 ± 2 CON: 22.1 ± 2	Creatine: NR CON: NR	Creatine monohydrate: 20 g/day	No loading	Without exercise training	Energy (kcal): Creatine: 2674.8 ± 879.6 CON: 2531.8 ± 519.5

Abbreviations: F, female; M, male; Exe, exercise; CON, control; kcal, kilocalorie; w, week; d, day, Min, minute; g, gram; kg, kilogram; h, hour; NR, not reported; reps, repetitions; RM, repetition maximum, RT, resistance training; HVRT, high-velocity resistance training; HIIT, high-intensity interval training; IU, international unit; FFM, free-fat mass; ST, strength training. For studies marked with identical age or BMI values between groups, these characteristics were reported for the total sample at baseline and not separately for the creatine and control groups.

## Data Availability

The original contributions presented in the study are included in the article, further inquiries can be directed to the corresponding author.

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
