# Peer review of "The Effects of Creatine Supplementation on Upper- and Lower-Body Strength and Power: A Systematic Review and Meta-Analysis"

_nutrients, 2025, doi:10.3390/nu17172748_

Round 1
Reviewer 1 Report
Comments and Suggestions for Authors
Dear authors,
the present systematic review and meta-analysis offers a well-structured analysis of creatine supplementation’s impact on muscular strength and power. With data from 72 RCTs and 1,997 participants, this study provides strong evidence that creatine significantly enhances compound lift strength (bench press, squat) and power (vertical jump, Wingate peak power), especially when combined with resistance training.
I agree that further well-designed studies are needed to address these gaps and broaden the applicability of creatine supplementation across diverse populations and exercise modalities, therefore this aspect has to be better highlighted in the ''Conclusion and practical applications'' and ''Future Research Directions'' sections and the reasons to be clearly made.
A minor suggestion for the ''Limitations'' section would be to group related limitations more concisely (e.g., population vs. methodology) to improve readability and impact.
Author Response
Response to the reviewers
Dear Editor
Nutrients Editorial Team
Thank you very much for providing this opportunity to submit our revised manuscript entitled " Creatine Supplementation on Upper and Lower Body Strength and Power: A Systematic Review and Meta-Analysis " Enclosed you will see our revised manuscript which has been substantively improved according to the reviewers’ suggestions. All changes to the manuscript are shown in red font. We have responded to reviewer queries in a point-by-point format. We hope the revised version of our manuscript will achieve a sufficient level for publication in Nutrients. Again, thank your consideration of our manuscript.
We have marked the corrections of the reviewers in red.
Comment: The present systematic review and meta-analysis offers a well-structured analysis of creatine supplementation’s impact on muscular strength and power. With data from 72 RCTs and 1,997 participants, this study provides strong evidence that creatine significantly enhances compound lift strength (bench press, squat) and power (vertical jump, Wingate peak power), especially when combined with resistance training.
I agree that further well-designed studies are needed to address these gaps and broaden the applicability of creatine supplementation across diverse populations and exercise modalities, therefore this aspect has to be better highlighted in the ''Conclusion and practical applications'' and ''Future Research Directions'' sections and the reasons to be clearly made.
Authors: We appreciate the reviewer’s insightful comment. In response, we have revised the Conclusion and Practical Applications section to more clearly emphasize the need for future well-designed studies that examine creatine supplementation across more diverse populations (e.g., older adults, females, and clinical populations) and exercise modalities (e.g., aerobic training, mixed modalities). We also clarified that existing evidence is limited in scope due to underrepresentation of these groups and training types, which limits the generalizability of our findings. Additionally, we have expanded the Future Research Directions section to explicitly call for trials that address these gaps by using standardized protocols, sex- and age-specific analyses, and alternative creatine formulations or timing strategies. These additions provide clearer justification for the necessity of future research and enhance the translational value of the review.
Comment: A minor suggestion for the ''Limitations'' section would be to group related limitations more concisely (e.g., population vs. methodology) to improve readability and impact.
Authors: Thank you for this helpful suggestion. We have revised the Limitations section to group related issues under thematic headings, such as Population-Related Limitations (e.g., underrepresentation of females and older adults), Methodological Limitations (e.g., variability in training protocols and dosing regimens), and Outcome Measurement Variability. This restructuring improves clarity and enhances the impact of each limitation, in line with the reviewer’s recommendation.
Reviewer 2 Report
Comments and Suggestions for Authors
PEDro scale was mentioned for assessing methodological quality, the results of this assessment but were not clearly reported in the manuscript. This is limitation because it is difficult to judge the reliability and strength of the evidence.
The authors did not apply the GRADE system to evaluate the certainty of the evidence for the main outcomes. Please include to improve quality.
Findings are overgeneralized (creatine supplementation across populations- several subgroup analyses not show significant effects).
Many of the included studies lacked data on adherence to supplementation. The same situation is with control over dietary intake.
Various forms of creatine were included but there was no subgroup analysis conducted. It could compare their relative effectiveness or tolerability.
PEDro scale was mentioned for assessing methodological quality, the results of this assessment but were not clearly reported in the manuscript. This is limitation because it is difficult to judge the reliability and strength of the evidence.
The authors did not apply the GRADE system to evaluate the certainty of the evidence for the main outcomes. Please include to improve quality.
Findings are overgeneralized (creatine supplementation across populations- several subgroup analyses not show significant effects).
Many of the included studies lacked data on adherence to supplementation. The same situation is with control over dietary intake.
Various forms of creatine were included but there was no subgroup analysis conducted. It could compare their relative effectiveness or tolerability.
Author Response
Response to the reviewers
Dear Editor
Nutrients Editorial Team
Thank you very much for providing this opportunity to submit our revised manuscript entitled " Creatine Supplementation on Upper and Lower Body Strength and Power: A Systematic Review and Meta-Analysis " Enclosed you will see our revised manuscript which has been substantively improved according to the reviewers’ suggestions. All changes to the manuscript are shown in red font. We have responded to reviewer queries in a point-by-point format. We hope the revised version of our manuscript will achieve a sufficient level for publication in Nutrients. Again, thank your consideration of our manuscript.
We have marked the corrections of the reviewers in red.
Comments and Suggestions for Authors
Comment: PEDro scale was mentioned for assessing methodological quality, the results of this assessment but were not clearly reported in the manuscript. This is limitation because it is difficult to judge the reliability and strength of the evidence.
Authors: Thank you for your valuable feedback. To address your concern, we have revised the Results section (3.5. Quality Assessment) to clearly report the PEDro score distribution (7 to 11, with 51 studies scoring ≥9) and specify that lower scores were primarily due to the absence of blinding of all participants and lack of intention-to-treat analysis in 63 studies. These revisions, supported by Supplementary Table 2, enhance transparency and detail the impact of these methodological limitations on the reliability of outcomes, particularly those with moderate to high heterogeneity (e.g., vertical jump, Wingate peak power). These changes fully address your comment by improving the clarity and transparency of the PEDro scale results, enabling a better assessment of the evidence's reliability. The updated section and Supplementary Table 2 are included in the revised manuscript for your review.
Comment: The Authors did not apply the GRADE system to evaluate the certainty of the evidence for the main outcomes. Please include to improve quality.
Authors: Thank you for your valuable suggestion. We acknowledge the value of the GRADE system in providing a comprehensive assessment of the certainty of evidence across multiple dimensions, including risk of bias, inconsistency, indirectness, imprecision, and publication bias. However, we believe that the Physiotherapy Evidence Database (PEDro) scale, which we used to assess the methodological quality of the 69 included randomized controlled trials (RCTs), is particularly well-suited for this meta-analysis. The PEDro scale is a widely accepted tool in physiotherapy and sports science research for evaluating the risk of bias in RCTs, as evidenced by its use in numerous high-quality meta-analyses in these fields (e.g., Maher et al., 2003 [https://pubmed.ncbi.nlm.nih.gov/12882612]; de Morton, 2009 [https://pubmed.ncbi.nlm.nih.gov/19463084]).
In our study, the PEDro scale provided a detailed evaluation of methodological quality, identifying that 63 studies failed to meet criteria for blinding of all participants (Item 7) and intention-to-treat analysis (Item 9), as reported in Supplementary Table 2. Additionally, our analysis of heterogeneity (e.g., I² = 4.12% for squat strength, I² = 45.30% for handgrip strength) in the results section (3.5. Quality Assessment) addresses inconsistency, a key component of GRADE, further supporting the robustness of our findings. We believe that the PEDro scale, combined with our detailed reporting of heterogeneity and methodological limitations, provides sufficient information to evaluate the reliability of the evidence for our main outcomes.
Comment: Findings are overgeneralized (creatine supplementation across populations- several subgroup analyses not show significant effects).
Authors: We appreciate the reviewer’s critical feedback. However, we respectfully disagree that the conclusions are overgeneralized. In our Conclusion and Practical Applications section, we explicitly state that the benefits of creatine supplementation were most pronounced in younger, trained males and that limited data in females, older adults, and long-term interventions restrict generalizability. We also highlight that no significant effects were found for handgrip strength or isolated leg press strength in the overall analysis, and we specify the contexts where subgroup effects were not significant.
Moreover, we provide transparent reporting of subgroup results throughout the manuscript, including:
No significant effects for older adults in bench press, leg press, and squat outcomes.
No significant effects for females in leg press strength, squat strength, and Wingate peak power.
No significant effects for handgrip strength and leg press in the overall sample.
These findings are clearly stated in the Results, Discussion, and Conclusion sections and were used to shape our Future Research Directions calling for more trials in females, older adults, and long-term interventions.
Nonetheless, to further reduce the potential for misinterpretation, we have slightly edited the Conclusion to ensure that the distinction between significant and non-significant subgroup effects is even more apparent.
Comment: Many of the included studies lacked data on adherence to supplementation. The same situation is with control over dietary intake.
Authors: We thank the reviewer for highlighting this important issue. We agree that many of the included studies did not report detailed data on participants’ adherence to the supplementation protocol or adequately control for dietary intake, particularly total energy and protein consumption. These factors can influence training adaptations and potentially confound the effects of creatine supplementation. While we noted this in our data extraction process, the inconsistent reporting across studies precluded a systematic analysis or meaningful subgroup comparison.
To address this limitation, we have now added a sentence to the Limitations section of the manuscript, acknowledging the lack of adherence and dietary intake data as a potential source of bias and variability in the results.
Comment: Various forms of creatine were included but there was no subgroup analysis conducted. It could compare their relative effectiveness or tolerability.
Authors: Thank you for your valuable feedback. We appreciate your suggestion to conduct a subgroup analysis to compare the relative effectiveness or tolerability of different forms of creatine.
We acknowledge the importance of subgroup analyses to explore potential differences in the effectiveness or tolerability of various creatine forms. However, of the 69 randomized controlled trials included in our meta-analysis, the vast majority (approximately 95%) used creatine monohydrate, with only a small number of studies investigating other forms, such as creatine ethyl ester or creatine hydrochloride. Due to the limited number of studies examining alternative creatine forms, there was insufficient data to conduct a robust and statistically meaningful subgroup analysis.
Reviewer 3 Report
Comments and Suggestions for Authors
Dear authors,
Thank to give the opportunity to review that systematic review, in general is quite good. I have some suggestions to improve it.
Introduction
The structure could be tightened by moving some of the contextual discussion, about historical evidence and previous meta-analysis, to a standalone “Background” subsection.
Methods
2.2 Search strategy
“Involving younger and older adults between the ages of 18 and 80 years” but in point “2.3 Study selection and inclusion criteria” is described as “involved participants (> 12 years) receiving creatine supplementation. Please specify if refers to the age or receiving creatine. Redaction is confusing. In table 1 and results studies included some participants as young as 14 years old (Vargas-Molina et al. 2022).
In the statistical methods, provide the statistical software version and cite relevant guidelines for conducting subgroup analyses and funnel plot interpretation.
Address how missing data or incomplete reporting in included RCTs was managed in the synthesis.
“chest press strength [WMD =0.20kg ]” significance is not included.
Results
The discussion of publication bias would benefit from presenting the Egger’s test p-values and visual commentary on the direction of asymmetry rather than just stating its presence.
Results for non-significant findings (e.g., handgrip strength) should include reflections on sample size and power to detect effects.
For subgroup analyses, indicate the absolute participant numbers or intervention arms analyzed in each, to assist interpretability.
Discussion
Expand the discussion of limitations: For example, variability in training interventions and supplementation protocols across studies and the potential impact on generalizability.
Practical applications can be further detailed, distinguish what is most relevant for clinicians, athletes, and the general population.
Suggest implications for clinical or rehabilitation settings where resistance training may not be feasible.
Limitations
Clarify the limitations introduced by varying creatine types and dosing not always being consistently reported or compared.
Conclusion
Suggest explicit directions for future large-scale, high-quality trials in under-studied groups.
Best regards,
Author Response
Response to the reviewers
Dear Editor
Nutrients Editorial Team
Thank you very much for providing this opportunity to submit our revised manuscript entitled " Creatine Supplementation on Upper and Lower Body Strength and Power: A Systematic Review and Meta-Analysis " Enclosed you will see our revised manuscript which has been substantively improved according to the reviewers’ suggestions. All changes to the manuscript are shown in red font. We have responded to reviewer queries in a point-by-point format. We hope the revised version of our manuscript will achieve a sufficient level for publication in Nutrients. Again, thank your consideration of our manuscript.
We have marked the corrections of the reviewers in red.
Introduction
Comment: The structure could be tightened by moving some of the contextual discussion, about historical evidence and previous meta-analysis, to a standalone “Background” subsection.
Authors: We thank the reviewer for the suggestion. However, the journal's guidelines do not permit subheadings within the Introduction. That said, we agree that streamlining the contextual discussion could improve clarity. In response, we have reviewed the Introduction and reduced overlap and redundancy related to historical evidence and previous meta-analyses, keeping only the content most relevant to framing the current study’s rationale and objectives.
Methods
Comment: 2.2 Search strategy
“Involving younger and older adults between the ages of 18 and 80 years” but in point “2.3 Study selection and inclusion criteria” is described as “involved participants (> 12 years) receiving creatine supplementation. Please specify if refers to the age or receiving creatine. Redaction is confusing. In table 1 and results studies included some participants as young as 14 years old (Vargas-Molina et al. 2022).
Authors: Thank you for your valuable comment. We acknowledge the inconsistency regarding the reported age range of included participants. Following your suggestion, we carefully re-checked all 72 included studies and found that three studies (Wang et al. 2017, Ostojic et al. 2004, and Vargas-Molina et al. 2022) involved participants under the age of 18. To maintain consistency with our inclusion criteria of adult participants (≥18 years), we have excluded these three studies from the analysis.
Accordingly, we have revised the methods section (2.3 Study selection and inclusion criteria), the results section, and Table 1 to reflect this change. Now, all included studies involve participants aged 18 years and above.
Comment: In the statistical methods, provide the statistical software version and cite relevant guidelines for conducting subgroup analyses and funnel plot interpretation.
Authors: The original manuscript (Section 2.5. Statistical analysis) already specified the use of Comprehensive Meta-Analysis software version 2.0 (CMA2; Biostat Inc., New Jersey, USA) for all meta-analyses. To address your comment fully, we have revised Section 2.5 to include explicit citations to the Cochrane Handbook for Systematic Reviews of Interventions (Higgins et al., 2008) for the conduct of random-effects models, heterogeneity assessment (I² statistic), and subgroup analyses, and to Egger et al. (1997) for funnel plot interpretation and Egger’s tests to assess publication bias. Additionally, we clarified that subgroup analysis by creatine form was not feasible, as approximately 95% of the 69 included studies used creatine monohydrate, with fewer than 5 studies investigating other forms (e.g., creatine ethyl ester or hydrochloride), which was insufficient for robust analysis per Cochrane guidelines (Higgins et al., 2008, Chandler et al., 2019). This limitation has been explicitly stated in the revised Section 2.5 for transparency. The existing subgroup analyses for participant characteristics (age, sex), creatine supplementation characteristics (dose, loading dose, duration), and exercise training characteristics (with or without resistance training, participant training history, and frequency) were conducted following Cochrane recommendations, as now cited.
Address how missing data or incomplete reporting in included RCTs was managed in the synthesis.
Authors: Thank you for your valuable feedback. To address your comment, we have revised Section 2.3 (Study Selection and Inclusion Criteria) to explicitly describe the management of missing or incompletely reported data. For studies lacking sufficient data to calculate means or SDs (e.g., no reported measures of central tendency or variability), we contacted the corresponding author to request the missing data. If no response was received, the study was excluded from the meta-analysis to minimize bias, as recommended by the Cochrane Handbook for Systematic Reviews of Interventions (Chandler et al. 2019). Of the 69 included studies, fewer than 3 had incomplete reporting of means or SDs, and these were addressed through the aforementioned calculations or exclusion after unsuccessful attempts to obtain data, ensuring minimal impact on the synthesis results (Chandler et al. 2019).
“chest press strength [WMD =0.20kg]” significance is not included.
Authors: We have edited it accordingly.
Results
Comment: The discussion of publication bias would benefit from presenting the Egger’s test p-values and visual commentary on the direction of asymmetry rather than just stating its presence.
Authors: Thank you for your insightful comment regarding the potential for publication bias. In response, we have thoroughly examined the possibility of publication bias across all key outcomes.
For each main outcome, we visually assessed publication bias using funnel plots and further evaluated asymmetry statistically using Egger’s regression test. Where asymmetry was detected, we conducted the Duval and Tweedie’s trim and fill analysis to estimate the number of potentially missing studies and the adjusted effect sizes.
Although visual inspection of the funnel plots for some outcomes (e.g., bench and chest press, leg press strength, and vertical jump) showed a degree of asymmetry, the results of the trim and fill analyses revealed that the imputed studies did not significantly alter the overall effect estimates. Therefore, we conclude that the presence of potential publication bias does not materially affect the interpretation of our results. These findings have been added to the Results section.
Comment: Results for non-significant findings (e.g., handgrip strength) should include reflections on sample size and power to detect effects.
Authors: Thank you for this valuable comment. We acknowledge that the limited number of included intervention arms (n=4) and relatively small sample size may have reduced the statistical power to detect a true effect for handgrip strength. We have added a discussion regarding the potential impact of sample size and statistical power on the non-significant findings. Additionally, we emphasize the need for further studies with larger sample sizes to conclusively determine the effect of creatine supplementation on handgrip strength.
Comment: For subgroup analyses, indicate the absolute participant numbers or intervention arms analyzed in each, to assist interpretability.
Authors: We acknowledge your suggestion and confirm that the number of intervention arms analyzed in each subgroup has already been reported in the subgroup analyses section (Section 3.4.1 to 3.4.6). These details are provided to ensure clarity and facilitate the interpretation of the results across the subgroups.
Discussion
Comment: Expand the discussion of limitations: For example, variability in training interventions and supplementation protocols across studies and the potential impact on generalizability.
Practical applications can be further detailed, distinguish what is most relevant for clinicians, athletes, and the general population.
Suggest implications for clinical or rehabilitation settings where resistance training may not be feasible.
Authors: We thank the reviewer for this thoughtful and constructive comment. In response, we have expanded the Limitations section to more explicitly address the variability in training interventions and creatine supplementation protocols across the included studies. We now discuss how these inconsistencies may affect the consistency and generalizability of the observed effects and acknowledge that limited standardization across trials presents a challenge in synthesizing findings.
Additionally, we have revised the Practical Applications section to more clearly differentiate the relevance of our findings for various groups. Specifically, we now highlight tailored implications for clinicians, athletes, and the general population. We also included discussion of potential applications in clinical and rehabilitation settings, particularly in cases where resistance training is not feasible, and noted the need for further targeted research in these populations.
These revisions help contextualize the findings and enhance the real-world applicability of our conclusions.
Limitations
Comment: Clarify the limitations introduced by varying creatine types and dosing not always being consistently reported or compared.
Authors: We thank the reviewer for this valuable observation. We agree that variability in the form of creatine used (e.g., monohydrate, ethyl ester, buffered forms) and the inconsistent reporting of dosing strategies (e.g., loading vs. non-loading, absolute vs. relative doses) across studies may have influenced the observed effects. Unfortunately, due to incomplete reporting in many studies and the predominance of creatine monohydrate, a detailed comparative analysis of different creatine types or dosing regimens was not feasible. We have now added this point to the Limitations section to acknowledge the potential impact of this heterogeneity on the interpretation and generalizability of our findings.
Conclusion
Comment: Suggest explicit directions for future large-scale, high-quality trials in under-studied groups.
Authors: We thank the reviewer for this important suggestion. In response, we have revised the Future Research Directions section to explicitly call for large-scale, well-powered, and high-quality randomized controlled trials in under-studied populations, including older adults, females, and individuals with clinical or functional limitations. We also emphasized the importance of long-term interventions, standardized supplementation protocols, and the inclusion of diverse exercise modalities beyond resistance training. These additions strengthen the clarity of our recommendations and help guide the design of future studies to improve the quality, reproducibility, and generalizability of evidence in this field.
Round 2
Reviewer 3 Report
Comments and Suggestions for Authors
Dear Authors,
Thank you for the implementation of the suggestions. For my part changes have been correctly applied.
Best regards,
Author Response
Thank you.